# Simulating the influence of snow surface processes on soil moisture dynamics and streamflow generation in an Alpine catchment

Nander Wever[1,2], Francesco Comola[1], Mathias Bavay[2], and Michael Lehning[2,1]

1 École Polytechnique Fédérale de Lausanne (EPFL), School of Architecture, Civil and Environmental Engineering, Lausanne, Switzerland.

2 WSL Institute for Snow and Avalanche Research SLF, Davos, Switzerland.

*Correspondence to:* Nander Wever (wever@slf.ch)

**Abstract.** The assessment of flood risks in alpine, snow covered catchments requires an understanding of the linkage between the snow cover, soil and discharge in the stream network. Here, we apply the comprehensive, distributed model Alpine3D to investigate the role of soil moisture in the predisposition of the Dischma catchment in Switzerland to high flows from rainfall and snowmelt. The recently updated soil module of the physics based, multi-layer snow cover model SNOWPACK, which solves the surface energy and mass balance in Alpine3D, is verified against soil moisture measurements at seven sites and various depths inside and in close proximity to the Dischma catchment. Measurements and simulations in such terrain are difficult and consequently, soil moisture was simulated with varying degrees of success. Differences between simulated and measured soil moisture mainly arises from an overestimation of soil freezing and an absence of a groundwater description in the Alpine3D model. Both were found to have an influence in the soil moisture measurements. Using the Alpine3D simulation as the surface scheme for a spatially-explicit hydrologic response model using a travel time distribution approach for interflow and baseflow, streamflow simulations were performed for the discharge from the catchment. The streamflow simulations provided a closer agreement with observed streamflow when driving the hydrologic response model with soil water fluxes at 30 cm depth in the Alpine3D model. Performance decreased when using the 2 cm soil water flux, thereby mostly ignoring soil processes. This illustrates that the role of soil moisture is important to take into account when understanding the relationship between both snowpack runoff and rainfall and catchment discharge in high alpine terrain. However, using the soil water flux at 60 cm depth to drive the hydrologic response model also decreased its performance, indicating that an optimal soil depth to include in surface simulations exists and that the runoff dynamics are controlled by only a shallow soil layer. Runoff coefficients (i.e., ratio of rainfall over discharge) based on measurements for high rainfall and snowmelt events were found to be dependent on the simulated initial soil moisture state at the onset of an event, further illustrating the important role of soil moisture for the hydrological processes in the catchment. The runoff coefficients using simulated discharge were found to reproduce this dependency which shows that the Alpine3D model framework can be successfully applied to assess the predisposition of the catchment to flood risks from both snowmelt and rainfall events.

# 1 Introduction

Alpine catchments are sensitive to flooding events (*Frei et al.*, 2000), with positive contributing factors being, for example, the topography, high rainfall rates and shallow soil depths (*Weingartner et al.*, 2003). The presence of a snow cover, acting as a water storage over winter, may dampen flood risks during some parts of the year (*Weingartner et al.*, 2003), but also provides
an important contribution to catchment scale runoff via meltwater in spring. Correct estimations of snow cover and snowmelt distributions are therefore essential for accurate streamflow simulations (*Maurer and Lettenmaier*, 2003; *Berg and Mulroy*, 2006; *Seyfried et al.*, 2009; *Koster et al.*, 2010). Additionally, rain-on-snow events may significantly increase the liquid water outflow from the snowpack (*Mazurkiewicz et al.*, 2008; *Wever et al.*, 2014a; *Würzer et al.*, 2016, 2017) and many flooding events have been caused by such events (*Marks et al.*, 2001; *Rössler et al.*, 2014).

However, accurate simulations of liquid water draining from the snowpack due to snowmelt or rainfall (henceforth termed snowpack runoff) are not sufficient to understand catchment runoff. The degree of saturation of the soil was found to determine the eventual effect of snowpack runoff on streamflow (*McNamara et al.*, 2005; *Seyfried et al.*, 2009; *Bales et al.*, 2011). This effect is not limited to snowpack runoff, but is also found for rainfall (*Bales et al.*, 2011; *Penna et al.*, 2011). During the winter months, the snow cover basically decouples the soil from the atmosphere and the upper boundary for the soil is determined by
the state of the snow cover on top (*McNamara et al.*, 2005; *Kumar et al.*, 2013). Often, the hydrological processes are strongly reduced during winter time, such as groundwater flow and streamflow, until the spring snowmelt provides liquid water again to the hydrological system. A model system to assess the hydrologic response of a catchment is therefore required to simulate both the soil and the snowpack accurately.

To assess this coupling between snowmelt, soil moisture and streamflow, the use of physics based models of snow surface
process descriptions in hydrological models seems attractive as they should not require calibration for the specific application. For example, *Rigon et al.* (2006) show that the physics based hydrological model GEOtop, which includes a relatively simple physics based snow scheme, is able to provide accurate streamflow simulations for small catchments, where a snow cover is present for extended periods during the winter season. *Kumar et al.* (2013) also found that using a physics based model approach for snow related processes in the PIHM model achieved a slightly better performance for streamflow simulations
than a temperature index approach. The results in their study suggest that this improvement is linked to the spatial variability of snow distribution and snowmelt, which provides a strong control on other components of the hydrological cycle, like soil moisture or streamflow. In *Warscher et al.* (2013), a similar comparison was made by comparing a temperature-index approach with an energy balance approach to determine snowmelt in the physics based hydrological model WaSiM-ETH. Their results show that the energy balance approach provides improvements particularly at the small spatial scales typical of high alpine
headwater catchments. However, the improvements rapidly decrease with increasing scale. It has been argued that simple temperature index based snowmelt models may perform well after careful calibration (*Kumar et al.*, 2013; *Comola et al.*, 2015a) and those models are still commonly used in operational flood forecasting. Nevertheless, physics based snow models may be considered more reliable when extrapolating to other conditions such as for climate change scenarios (e.g., *Bavay et al.* (2013)) or to catchments where limited calibration data is available.

The fully-distributed Alpine3D model is typically applied for detailed studies of small scale surface processes in alpine catchments where snow plays an important role (*Lehning et al.*, 2006; *Mott et al.*, 2008; *Groot Zwaaftink et al.*, 2013). In alpine terrain, considering the length scales less than a few 100 m is important as on these scales, wind drifts determine the snow accumulation and local topography heavily influences the energy balance via the slope aspect, angle and local shading.

In this study, the recent addition to the SNOWPACK model of a solver for Richards Equation for soil (see *Wever et al.* (2014a, 2015)) is verified against soil moisture measurements in the vicinity of Davos, Switzerland. The SNOWPACK model provides the surface scheme in the Alpine3D model framework, using physics based descriptions of soil-snow-vegetation processes (*Gouttevin et al.*, 2015). Here, the capabilities of Alpine3D to capture the soil moisture state is assessed. Furthermore, the Alpine3D model provides the surface scheme for a travel time distribution hydrologic response model to simulate catchment

discharge (*Comola et al.*, 2015b; *Gallice et al.*, 2016) and here the role of soil moisture in the coupling of Alpine3D to the hydrologic response model, as well as the influence of the soil moisture state on streamflow generation in the catchment is investigated.

## 2 Study Area and Data

### 2.1 Study Area

The Davos area is located in the Canton Grisons in east Switzerland. The studied area is defined as an area of $21.5 \times 21.5$ km$^2$ and stretches over an elevation range from about 1250 m above sea level (a.s.l.) to 3218 m a.s.l. Some small glaciers exist in the highest parts, covering about 0.86 km$^2$ (*Zappa et al.*, 2003). The Dischma catchment is an unregulated catchment of 43.3 km$^2$ in the Davos area and has been subject to previous studies concerning streamflow from the Dischma river (e.g., *Zappa et al.* (2003); *Lehning et al.* (2006); *Bavay et al.* (2009); *Schaefli et al.* (2014); *Comola et al.* (2015b)). The measurement site

Weissfluhjoch (WFJ), which is focussed on snow-related measurements, as well as several permanent meteorological stations are located in close proximity of the catchment. Figure 1 shows the studied area, including the measurement stations and the gauging station for streamflow measurements of the Dischmabach in the Dischma catchment. Quality controlled streamflow data, catchment properties and the border polygon have been provided by the Swiss Federal Office for the Environment (FOEN) (*Federal Office for the Environment (FOEN)*, 2015, 2017). Simulations presented in this study consist of three winter seasons,

from October 1, 2010 to September 30, 2013.

Snowfall plays an important role in the Davos area. Table 1 shows the precipitation sums for two heated rain gauges at two elevations in the region. About 40% to 80% of total precipitation falls as snow at the lower and upper parts of the Dischma catchment, respectively (*Zappa et al.*, 2003). Precipitation in the Davos area is commonly separated into rain and snowfall based on an air temperature threshold of 1.2 °C. The winter months are dominated by snowfall at all elevations in the area. In the

30 meteorological summer months (June-August), about 7% of the precipitation amounts still consist of snowfall at 2536 m a.s.l. At the lower rain gauge, almost all precipitation falls as rain in the meteorological summer months. The two precipitation gauges show a strong elevation gradient in precipitation: at 2536 m a.s.l., precipitation amounts are about 1.9 times higher than at 1590 m a.s.l. This elevation gradient may, however, overestimate the true areal-mean gradient because the upper site may

be limited representative for the Dischma catchment (*Wirz et al.*, 2011; *Grünewald and Lehning*, 2015). Furthermore, the area exhibits a climatological northwest - southeast gradient in precipitation (*Vögeli et al.*, 2016).

Figures **??** and **??** show the daily temperature and precipitation amounts separated in snowfall and rainfall, for both locations with a heated rain gauge. The yearly cycle in temperature has a similar amplitude at both elevations. Maximum daily temperatures occasionally surpassed $20°$C at 1590 m a.s.l. and $15°$C at 2536 m a.s.l. The minimum daily temperatures reached $−20°$C and $−25°$C, respectively. Note that those low temperatures were reached after significant snowfall in the months before. Therefore, the isolating snow cover is expected to have prevented an impact of these cold days on soil freezing.

An important event in the meteorological forcing can be found in winter season 2011-2012, which was dominated by large snowfalls in December, January and February. Maximum measured snow depth was higher than in the other simulated years. Cold temperatures in those months were followed by a relatively warm spring season, resulting in relatively high snowmelt rates. Also the spring of snow season 2010-2011 was relatively warm, compared to the spring of 2012-2013. None of the summer periods were outspokenly dry or wet, and precipitation occurred homogeneously distributed over time, with the exception of the dry November 2011, in which no precipitation occurred. Finally, total precipitation at WFJ in summer 2011 was similar to summer 2012, whereas the summer 2013 was rather dry in Davos.

## 2.2  Data

Several measurement sites are located or were temporarily installed in the vicinity of Davos. Their locations are shown in Figure 1. The sensitivity of Alpine3D simulations to input data coverage as well as specific interpolation and modelling choices is discussed in detail in *Schlögl et al.* (2016). Here we operate with a standard set-up as described below and distinguish between 5 types of meteorological stations (see Table 2).

IMIS stations are permanently installed operational meteorological stations in the Swiss Alps, especially focused on usage for avalanche warning (*Lehning et al.*, 1999). The stations measure at 7.5 m above the ground and receive regular maintenance and quality control. One exception is SLF2 in Davos-Dorf, which is used as a test station for new sensors or hardware. During the winter season 2011 and for a large part of winter season 2012, the relative humidity sensor was providing erroneous data due to a faulty test sensor. The IMIS stations provide a good spatial coverage of the common meteorological parameters, but due to limited energy availability, lack heated rain gauges to assess solid precipitation.

In the Davos area, two heated rain gauges are present, located at the SwissMetNet stations WFJ (2536 m a.s.l./2691 m a.s.l.) and Davos-Dorf (1590 m a.s.l.), operated by the Swiss Federal Office of Meteorology and Climatology (MeteoSwiss). These stations thereby provide, after applying an undercatch correction, relatively accurate measurements of solid precipitation in winter, in addition to high quality measurements of common meteorological parameters. For example, these stations also provide incoming shortwave and longwave radiation using ventilated and heated sensors to prevent riming and snow covering up the sensors. At WFJ, shortwave and longwave radiation sensors located at a local mountain peak of 2691 m a.s.l. were used in this study. These sensors experience almost no shadowing from surrounding mountain peaks. The WFJ measurement site at 2536 m a.s.l. is equipped with ventilated temperature and relative humidity sensors. Moreover, several backup sensors are present, allowing for filling data gaps.

The IRKIS and SensorScope stations were temporarily set up for this study. IRKIS stations are based on the IMIS design, but with a sensor height of 4.5 m. SensorScope stations (*Ingelrest et al.*, 2010) were installed in less accessible terrain to increase quantity and area covered by measurements. Operation of these type of stations in the harsh winter conditions appeared to be more difficult than expected and the sometimes hazardous locations of the measurement sites was hindering maintenance during the winter season. Due to several outages of the stations and broken sensors, the meteorological measurement series contain many gaps and are not used as input in this study. The IRKIS and SensorScope stations were additionally equipped with soil moisture sensors at 10, 30, 50 cm depth. At IRKIS stations and the Golf Course SensorScope station, soil moisture sensors were also installed at 80 and 120 cm depth. This is schematically illustrated in Fig. 3. At each depth, two sensors, labelled "(A)" and "(B)" here, were installed at approximately 50 cm distance. The IRKIS station SLF2 was using the IMIS station SLF2, but soil moisture sensors were installed in close vicinity. IRKIS stations report weather and soil moisture conditions at a time resolution of 10 minutes. SensorScope stations measure at a time resolution of 1 minute, sending their data using GPRS cell phone networks.

The choice for the soil moisture measurement sites is motivated by the availability of an accessible flat area and by possibly well representing the catchment soil types. The Grossalp and Pischa stations were located in the "alpine meadow" land use class, which is 21.1% of the land use coverage (see Table 3). The Uf den Chaiseren, Dorfji and Stillberg stations are located in the "mixed forest", "bush" and "bare soil" classes, respectively, which are found in 12.9%, 7.3% and 6.0% of the Dischma catchment, respectively. The SLF2 and Golf Course stations would officially fall into the category of "settlement", but one would describe the area as "alpine meadow".

At the soil moisture measurement sites, Decagon 10HS soil moisture sensors were installed, which have a volume of influence of 1320 mL, or a volume of approximately 11x11x11 cm (*Decagon Devices*, 2014). *Mittelbach et al.* (2012) present an in-depth comparison with other types of soil moisture sensors. A few important issues related to the Decagon 10HS sensors that are relevant for this study were reported. In their study, the liquid water content values from the sensors exhibited a soil temperature dependency. The sensors were also found to hardly register values above 0.40 m$^3$ m$^{-3}$ and it was concluded that the 10HS is showing a decreased sensitivity with increasing liquid water content. Consequently, the sensors are unable to follow fluctuations in wet soil conditions. For some of the sites and depths where we installed these type of sensors, the measured LWC is around or above 0.40 m$^3$ m$^{-3}$. We therefore expect a strongly reduced dynamic response in these locations. However, many of the installed sensors were recording values well below 0.40 m$^3$ m$^{-3}$ and provide useful measurements. The dielectric constant of ice is much lower than for water, making the sensors mostly sensible to the liquid water content part only.

## 3 Methods

### 3.1 Simulation Setup

SNOWPACK is a one-dimensional physics based multi-layer snow cover model (*Lehning et al.*, 2002a, b), which provides the surface scheme for Alpine3D. Richards equation (*Richards*, 1931) is used to describe soil moisture dynamics and numerically solved using finite differences scheme over the model layers (elements). Water flow in snow is solved by the bucket scheme,

which provides accurate snowpack runoff estimations on daily and seasonal time scales (*Wever et al.*, 2014b), and has notice-able lower computational costs (in the order of a factor 2-3) than using the full Richards equation for snow. The liquid water outflow from the snowpack is prescribed as the upper boundary condition for the Richards equation for the soil (*Wever et al.*, 2014b). In snow-free conditions, the upper boundary condition is defined by rainfall, evaporation and deposition resulting from the latent heat flux. Phase changes in soil are calculated following *Wever et al.* (2015). Water retention curves in the SNOW-PACK model are based on the van Genuchten model (*van Genuchten*, 1980) via predefined soil types as in the ROSETTA class average parameters (*Schaap et al.*, 2001).

To run simulations for the Dischma catchment, the Alpine3D model system is used, which describes surface processes in complex terrain by performing distributed SNOWPACK simulations (*Lehning et al.*, 2006). For describing the high spatial variability in incoming and outgoing long- and shortwave radiation, including shadowing effects and the surface reflections of shortwave radiation, a detailed energy balance module is available (*Michlmayr et al.*, 2008). An additional module considers drifting snow (*Lehning et al.*, 2008; *Mott et al.*, 2010), including sublimation processes (*Groot Zwaaftink et al.*, 2013). These drifting snow modules are not used in this study, as the location of the measurement sites are not prone to significant drifting snow effects, except for the Grossalp station. Moreover, the calculation of the wind fields and snow drift is posing a high computational demand compared to the other modules. The different modules and the coupling strategy is described in *Lehning et al.* (2006).

The Alpine3D simulations were run for a domain of 21.5 km×21.5 km with a grid cell size of 100 m×100 m, giving a total size of $215 \times 215$ grid cells. The model was run in hourly time steps, providing meteorological forcing data per time step for each pixel by interpolating from the meteorological stations in and just outside the Davos area using the MeteoIO library (*Bavay and Egger*, 2014). Per hourly time step, 4 SNOWPACK time steps are executed at 15 min. resolution.

At each Alpine3D model time step, the precipitation measurements from the heated rain gauges in Davos and WFJ were interpolated over the grid by using the elevation gradient from the measurements. The commonly used temperature threshold in the SNOWPACK model of 1.2 °C was used to separate precipitation into rain and snowfall. Air temperature, relative humidity and wind speed were also interpolated over the grid, using the station data as indicated in Table 2 and applying inverse distance weighting interpolation with lapse rates calculated from the available data. Only IMIS stations were used for spatial interpolations, except for the radiation components. Incoming longwave radiation was interpolated using a lapse rate between both SwissMetNet stations providing radiation. Shortwave radiation is provided by the radiation module, using the measurements from WFJ. The radiation balance is closed by the SNOWPACK simulations at each grid points, when SNOWPACK calculates the surface temperature and surface albedo.

Two important components to initialise Alpine3D simulations are the digital elevation model (DEM) and distributed soil information. For the Davos area, the DEM is provided by the Swiss Federal Office of Topography (swisstopo). Soil properties were based on the land use classification, as provided by swisstopo (*Zappa et al.*, 2003). Table 3 lists the land use classes, the percentage of areal coverage in the simulated area and the soil properties. Pixels that were defined as glacier, ice, firn, road, settlements, rivers and lakes (6%) were initialised in a state that represents the land use class. Other vegetation free areas are classified as rocky surface. This class is assigned to 29% of the pixels and consist for a large part of ground moraine and scree

slopes, whereas solid rock and rock walls are sparse in the Davos area. The rocky surface pixels were initialised uniformly with loamy sand. This is based on observations when installing soil temperature sensors at the WFJ, which is located in the rock class and for which plausible simulations were obtained using this soil class (*Wever et al.*, 2015). All other pixels (65%), including forests, meadows, pasture, bare soil, and occasional pixels that are defined as agricultural use were initialised

using an upper layer of 60 cm consisting of silt loam and a lower layer of 240 cm consisting of sandy loam. This choice is based on observations when installing the soil moisture sensors at the IRKIS and SensorScope stations. The soil permeability classification provided by the Swiss Federal Office for Agriculture (FOAG) shows generally high permeability in the area surrounding Davos, which confirms the choice for soil types with no clay content. To determine thermal properties of the soil, literature values were taken (Table 4). For thermal conductivity, a wide range of values is reported and a strong dependence

with water content is present. We used values corresponding to typical soil saturation values, based on work by *Ochsner et al.* (2001) and *Bachmann et al.* (2001). The skeleton fraction of the soil is largely unknown, and altough it may impact the soil hydraulic properties significantly (*Brakensiek and Rawls*, 1994) and thereby soil moisture and streamflow simulations (*Rössler and Löffler*, 2010), the SNOWPACK model currently does not support pedotransfer functions that take the skeleton fraction into account, and hence, it was neglected in our simulations.

A soil depth of 3 m was simulated, subdivided into 23 layers, as illustrated in Fig. 3. The layer spacing was 2 cm near the surface, increasing to 25 cm at 3 m depth. The densely spaced surface layers are necessary to describe the large gradients of temperature and moisture occurring in this region. The lower boundary condition at 3 m depth was set as a water table condition for the liquid water flow and as a constant upward geothermal heat flux of $0.06$ W m$^{-2}$ for the heat equation.

For the simulations, atmospheric stability was taking into account when calculating the turbulent heat fluxes, using the

modified Stearns correction (*Schlögl et al.*, 2017). The roughness length during the presence of a snow cover was defined to be $0.015$ m below 1900 m a.s.l. and $0.002$ m otherwise. This division is based on the generally rougher terrain below 1900 m, due to the presence of trees or large bushes, whereas above 1900 m, mainly meadows and scree fields are present. When pixels are snow free, they were assigned a roughness length of $0.02$ m.

Alpine3D has recently been extended with MPI support, allowing for the parallelisation of the distributed SNOWPACK and

energy balance simulations. Using 36 CPU cores from a HPC system consisting of in total 32 compute nodes with two 6-core AMD Opteron 2439, 2.8 GHz processors per compute node, the computation took on average 14 hours wall clock time for a single year, mainly depending on the snow depth in the winter season.

## 3.2   Analysis

The soil moisture measurements series were first cleaned from erroneous data, like negative values, or data from broken sensors

after visual inspection of the time series. Then, data was aggregated to hourly and daily time scales by calculating average soil moisture contents over the respective time spans. From the simulations, the modelled soil moisture values were extracted for each depth at which measurements were taken. The output resolution was 1 hour and daily values were calculated by averaging the hourly values.

As the area of Davos is dominated by snowfall in winter, a separation is made for yearly, summer and winter periods. The summer months are defined as the period from June through October. At the elevation of the soil moisture stations, snowfall episodes are almost absent in these months and the winter snow cover has melted completely by the beginning of June. The winter months are defined as the period from November through May, when a snow cover is present. Note that typically, the snow cover melts away in April or May at the stations and in those months, the soil moisture is expected to be strongly influenced by the snowmelt from the snowpack.

The streamflow from the Dischmabach is calculated using a spatially explicit and semi-distributed hydrologic response model that casts the soil moisture dynamics in a travel time distribution framework (*Comola et al.*, 2015b; *Gallice et al.*, 2016). Specifically, the model simulates hydrologic transport within sub-catchment soil compartments and stream network, identified through geomorphological analysis of the digital elevation model (*Tarboton*, 1997). An upper soil compartment is recharged by a water flux provided by the surface scheme of the Alpine3D model. Part of the outflow from the upper soil compartment generates interflow, which represents the fast hydrologic response. The remaining part recharges a lower soil compartment, where the slow groundwater flow in generated. However, it is a-priori not clear where to draw the boundary between the surface scheme and the hydrologic response model. To investigate this, we tested 3 scenarios by using the soil water flux at 2, 30 and 60 cm depth (see Fig. 3). This approach allows the Alpine3D model to run with a thick soil layer (3 m), easing the choice of lower boundary condition for the soil (geothermal heat flux and a water table). The 2 cm flux represents a case where almost all water input into the soil from both snowmelt as well as rainfall is directly routed using the hydrologic response model, while at the same time ensuring that evaporation is taken into account. It basically represents the case where soil is neglected for the discharge simulations. The simulations using the flux at 30 or 60 cm depth are performed to verify the sensitivity of the hydrologic response model to the thickness of the soil layers used in Alpine3D. The water flux at all grid points whose centerpoint is inside the polygons of the 55 sub-catchments is summed and provided to the hydrologic response model. The sub-catchments are determined by analyzing the digital elevation model (*Comola et al.*, 2015b).

It is noteworthy that the hydrological model is parsimonious in terms of calibration parameters, owing to the explicit analysis of the catchment's geomorphological complexity and the physically-based simulation of surface processes provided by alpine3D. In particular, the two compartments and the recharge rate in the travel time distribution approach of the hydrologic response model gives three parameters that require calibration: the average travel time of the upper and lower soil compartment (day) and the maximum recharge rate of the lower compartment from the upper compartment (mm day$^{-1}$). Here, all three approaches which define the input for the hydrologic response model are independently calibrated with measured discharge from October 2004 to September 2009, using Monte-Carlo simulations with 5000 repetitions. The best combination of coefficients was determined based on the highest Nash-Sutcliffe Model Efficiency (NSE) coefficient (*Nash and Sutcliffe*, 1970). The period from October 2009 - September 2014 was used for validation.

To analyse the effect of soil moisture on streamflow generation, we calculated the average soil saturation in the top 40 cm of all pixels inside the Dischma catchment. This is the approximate depth which is captured by the volume of influence of the soil moisture sensors at 10 and 30 cm depths. Furthermore, we defined rainfall events as events for which the moving 12 hour sum exceeds 10 mm. The time series for the event selection was determined by taking the average value of both heated rain gauges.

The start of an event is defined as the first time step for which precipitation is present, and the end was determined when the cumulative 12 hour sum fell below 3 mm, after first reaching 10 mm. A similar approach was done for snowpack runoff from the model, where snowpack runoff is considered analogue to rainfall. With this procedure, in total 168 rainfall events and 301 snowpack runoff events were selected (i.e, on average 16.8 and 30.1 events per year, respectively). The average duration of an event is 21.8 hrs (rainfall) and 20.9 hrs (snowpack runoff). On average, there are 6.8 days in between rainfall events, excluding the winter season. There are 1.3 days in between snowpack runoff events, excluding the summer season, showing that these events are concentrated in the spring season.

## 4    Results and Discussion

### 4.1    Snow Height

Figure 4 shows measured and simulated snow depth by Alpine3D for stations SLF2, Uf den Chaiseren and Grossalp. In snow seasons 2011 and 2013, the snow depth in the Alpine3D simulations is satisfyingly reproduced at both SLF2 and Uf den Chaiseren. The snow depth at Grossalp is overestimated in all snow seasons. This is explained by the fact that this particular site is relatively sensitive to wind eroding snow from the surface. The snow depth in snow season 2012 is overestimated at all stations, which is related to unusual meteorological circumstances of large snowfalls accompanied by strong winds, which lead to an overestimation of precipitation as measured by the heated rain gauge (also discussed in *Wever et al.* (2015)). Nevertheless, the snow cover development at those three sites is overall satisfactorily simulated in Alpine3D for providing an upper boundary for the soil. In the summer months, grass growth below the sensor is visible as an increase in snow depth with a highly noisy signal. Mowing activity is indicated by sudden decreases in snow depth.

### 4.2    Soil Moisture

Figures 5 and 6 show measured and simulated soil moisture time series at all depths for 2 of the 7 stations in the area of Davos. Similar figures for the other 5 stations can be found in the Online Supplement. Temporal variations in soil moisture in the area of Davos are clearly dominated by winter periods, in which the presence of a snow cover reduces or inhibits water influx at the top of the soil for several months. This phase is followed by the snowmelt phase in spring, when liquid water draining from the snowpack is providing liquid water again to the soil. This is illustrated by way of example in Figures 7a,b for the SLF2 measurement site for the snowmelt season 2011. The onset of wetting of the soil due to snowmelt is well predicted. It illustrates that using the bucket scheme for water flow in snow is justified on daily and seasonal time scales (*Wever et al.*, 2014b). The diurnal cycle of snowmelt is also visible as a diurnal cycle on soil moisture levels, well reproduced by the simulations. The summer months are generally snow-free (Figures 5 and 6), and soil moisture measurements show fluctuations on short time scales of a few days, related to rainfall and evaporation. A detailed example hereof is shown in Figures 7c,d for the snow-free month June 2011 for the SLF2 measurement site. Particularly large rainfall events are strongly influencing soil moisture,

compared to small ones. Generally, the simulated soil moisture reacts more strongly to incoming rainwater and is also showing stronger fluctuations on sub-daily time scales than in the measurements.

At several stations, soil freezing is indicated by the soil moisture sensors. Significant soil freezing was occurring in snow season 2011, as clearly visible at SLF2 (Figure 5) and Uf den Chaiseren (Figure 6), as well as Stillberg (see Figure S2 in the Online Supplement). The soil freezing was promoted by a long period with no snow or only a shallow snow cover, allowing the soil to cool. For the stations SLF2 and Uf den Chaiseren, the onset of the freezing is rather well predicted in the Alpine3D simulations. At most stations, the soil freezing front does not seem to reach the sensor at 30 cm depth. Only at Uf den Chaiseren and Stillberg, the minimum soil moisture at this depth in this particular snow season is slightly lower than in the other snow seasons, which may be indicative of slight soil freezing here.

The simulations show soil freezing at 10 cm depth in all snow seasons at most stations, for at least a short period of time, which is more soil freezing than captured in the soil moisture measurements. The overestimation of soil freezing in the simulations may be partly related to neglecting the presence of vegetation at the measurement sites. All sites are covered by grass, or rough pasture and bushes. To account for the insulating effects of the canopy, some soil freezing schemes consider the presence of a canopy when calculating soil phase changes (e.g., *Giard and Bazile* (2000)). Due to the lack of possible validation data, we did not implement this. Furthermore, the amount of soil freezing is also dependent on the amount of liquid water available. At stations Grossalp and Golf Course, the soil is wetter than simulated, which would require a higher heat flow out of the soil before freezing may start and uncertainties in soil thermal properties may also play a role here. Finally, the neglectance of the skeleton fraction in our simulations could lead to an overestimation of hydraulic conductivity and introduce a negative bias in soil moisture (*Brakensiek and Rawls*, 1994). However, as discussed by *Rössler and Löffler* (2010), the spatial variability of the skeleton fraction is generally unknown. For example, for some sites we get an adequate soil moisture simulation without considering the skeleton fraction, whereas for other sites the simulations are showing less agreement with measurements. This would then only allow for an ad-hoc modification of the skeleton fraction, as we cannot separate well enough between the soil moisture sites based on available information (land use and soil permeability).

The relatively dry summer of 2013, most pronounced at low elevations as indicated by the difference in summer precipitation from both heated rain gauges (Table 1), is clearly visible in the simulations by a drop in soil moisture at all depths, reaching the lowest values of the entire measurement period. Unfortunately, soil moisture sensors had stopped working at many stations by this time, but at the site SLF2 and Stillberg, a good correspondence is found in the 10 cm measured and simulated soil moisture series. At the Uf den Chaiseren site, the recession curve in this summer is particularly present at the sensors at 50 and 80 cm depth, and absent in the highest sensor.

Some features are found that likely relate to hydrological processes that are not simulated in the Alpine3D model. For example, at the Uf den Chaiseren site, the soil moisture at 80 and 120 cm is clearly influenced by a rising water table in the late snowmelt season. This is indicated by the sudden rise to high values of saturation, remaining constant afterwards (Figure 6). The soil at the Golf Course station appeared to be close to saturation for extended periods of time (see Figure S5 in the Online Supplement), which is congruent with observations when installing the sensors. The location of these two stations close to the Dischmabach (Uf den Chaiseren) and Landwasser river (Golf Course), which are partly fed by meltwater from the glacierised

area, supports this interpretation. The apparent interaction with groundwater levels at these stations is not considered in the simulations, as the groundwater table is fixed at the lower boundary of the soil column in the model domain. Similarly, the measurements at 10 and 30 cm depth at the Grossalp station (see Figure S1 in the Online Supplement) also indicate high saturation of the soil, for which no source of water could be found. Due to the insensitivity of the soil moisture sensors in wet soil conditions, discrepancies between simulations and measurements as found at the sites Grossalp and Golf Course can only be assessed qualitatively and provide insights on the limitations of the measurements and simulations. In contrast with the other measurement sites, the soil moisture sensors at the Pischa station show a very dynamic response (see Figure S3 in the Online Supplement). We cannot exclude that during the installation of the sensors, the soil was disturbed in such a way that afterwards, efficient preferential flow paths occurred along the boundaries of the displaced soil layers.

Figure 8 shows the $r^2$ values between daily averaged measured and simulated soil moisture values for the various depths for the full period and for the summer months only. Here, soil moisture was taken as the sum of ice and water to compensate for the overestimation of soil freezing. Only the values for the sensor with the highest $r^2$ value of the two sensors per depth are shown. Generally, the highest $r^2$ is achieved for 30 cm and 50 cm depth. Closer to the surface, the overestimation in soil freezing, as well as the generally large gradients in soil moisture reduces the agreement. For deeper layers, groundwater dynamics as discussed above, which is not considered by the model, could be identified as contributing to lower model agreement. Results for the summer months show higher $r^2$ values for the 10 cm and 30 cm soil moisture sensors. These layers are particularly influenced by rainfall in these months, for which timing is more accurate in the model than the onset of snowpack runoff which determines soil moisture fluctuations in large periods of the year. For deeper layers, the model performance is comparable to the performance for the full year.

The $r^2$ values indicate that for many sites, some of the variability in soil moisture is adequately captured. This spatially varying reproducibility of soil moisture is a typical result for physics-based soil models applied in alpine terrain, as for example found also by *Rössler and Löffler* (2010); *Kumar et al.* (2013); *Pasolli et al.* (2013). Better agreement ($r^2$ between 0.8 and 0.95) may be achieved by calibrating the water retention curves, or related soil parameters, to the soil moisture measurements (*Gurtz et al.*, 2003; *Brocca et al.*, 2013; *Pellet et al.*, 2016), although the lack of distributed soil information would make a distribution of this calibration over the model domain difficult and not very meaningful.

### 4.3 Streamflow

Figure 9 shows the measured and simulated streamflow at the outlet of the Dischmabach in the Dischma catchment. The winter periods are clearly identifiable by the hydrograph falling back to baseflow. Furthermore, high discharge is particularly found in spring, during the snowmelt season which typically lasts from April to June in the Dischma catchment (*Griessinger et al.*, 2016). During the summer period, streamflow slowly decreases, interrupted regularly with peaks in streamflow due to rainfall. These general discharge patterns are well captured in the simulations, regardless of the depth below the surface where the liquid water flux is routed to the runoff model. However, the fast dynamics on daily time scales in the Dischmabach streamflow is underestimated in the simulations, particularly when using the flux at 60 cm depth, for which the deep soil layer apparently has a too strong dampening of the incoming water in order to reproduce daily streamflow behaviour. Improvements in reproducing

the dynamic response on short time scales in the simulations could probably be obtained by including lateral water transport in Alpine3D, which would allow to account for the fast surface runoff, which for example takes place over highly saturated or impervious soils.

The three simulations of streamflow differ in the depth at which the soil water flux was used for the travel time distribution approach. Figure 10 displays the NSE coefficients per year as well as the average for these three depths based on daily discharge. For the full validation period, the NSE coefficients for either the 2 cm, 30 cm or 60 cm flux show very similar scores of around 0.8. When the calculation of NSE coefficients is limited to the snow melt season (April-June) or the summer season (June-October) only, differences become more pronounced. Highest NSE coefficient is achieved with the flux at 30 cm depth. The results suggest that the updated soil module of SNOWPACK is enabling a good prediction of streamflow in the summer months. Interpreting the flux at 2 cm depth as the effect of routing snowpack runoff and rainfall minus evaporation to the hydrologic response model, it shows that including 30 cm of soil layers improves the discharge simulation.

We hypothesize that in the Dischma catchment, the snow melt season is providing large water fluxes from the snow to the soil, compared to the soil water dynamics, making it the dominant factor in predicting stream flow. In the summer months, however, the predisposition of the soil is also an important factor, thus neglecting the soil layers almost completely, by routing the 2 cm flux to the runoff model, is reducing the model efficiency. The improved results of the 30 cm soil simulations as opposed to using much deeper soils suggests that the temporal dynamics of near-surface water fluxes exert a relevant control on the hydrologic response of these Alpine catchments.

## 4.4 Predisposition from Soil Moisture

The soil moisture state of the Dischma catchment is summarized as the basin wide average saturation in the upper 40 cm of the soil at the onset of a rainfall or snowpack runoff event. The water flux at this depth provided the highest skill in reproducing observed discharge after applying the hydrologic response model. Figure 11a shows the runoff coefficient (i.e., the ratio of rainfall to discharge) for the cumulative rainfall and measured discharge from the Dischma catchment as a function of catchment average soil saturation. The figure illustrates that the reduced storage capacity in wetter soils leads indeed to more of the precipitation water being routed to discharge and vice versa. In Figure 11b, it is illustrated that similar behaviour is also captured in the simulated discharge. For the Dischma catchment, we found that not only the total event runoff coefficient is determined by the soil moisture state, but also the peak runoff coefficient, defined as the ratio of the maximum peak in precipitation over the maximum, not necessarily simultaneous, discharge peak (see Fig. 11c for measured discharge). This relationship is again also found for the simulated discharge (Fig. 11d). Although the initial soil moisture is impacting the runoff coefficient for both the cumulative amounts as well as the peak values, the time lag between a peak in rainfall and measured discharge is not dependent on the soil moisture conditions (Fig. 11e). Also this result is reproduced by the simulated discharge (see Fig. 11f). All $r^2$ values reported in Fig. 11 test significant at the 95 % confidence level.

When the catchment is snow-covered, the melt water outflow from the snowpack can be considered analoguous to rainfall in summer. A similar analysis as presented in Fig. 11 is performed using snowpack runoff (see Fig. 12). Also here we find that the soil moisture state at the onset of snowpack runoff events influences the streamflow discharge. Similar to rainfall events,

the soil moisture state influences the ratio of the cumulative measured event discharge over cumulative snowpack runoff (Fig. 12a) as well as the peak ratio (Fig. 12c). The correlation coefficients are higher for the snowpack runoff events than for the rainfall events. This higher correlation coefficient for snowpack runoff than rainfall is also found for the runoff coefficients using simulated discharge (Fig. 12b and d). Similar to rainfall events, the time delay between peaks in snowpack runoff and discharge is independent of the initial soil moisture state.

In line with previous studies (*Maurer and Lettenmaier*, 2003; *Berg and Mulroy*, 2006; *Seyfried et al.*, 2009; *Koster et al.*, 2010), the results confirm that the simulations of the soil moisture state contribute to the understanding of how rainfall and snowpack runoff input in the hydrological system is influencing discharge from the catchment. Based on measurements, this relationship was found for alpine catchments for summer rainfall (*Penna et al.*, 2011). However, we show that this effect is reproduced in both measured runoff coefficients as well as simulated ones and also exists for snowpack runoff. The relationship between the initial soil moisture state and runoff coefficients is similar for observed and simulated discharge as well as for rainfall or snowpack runoff events. These results suggest that simulations of soil moisture in snow dominated catchments using the Alpine3D model combined with the hydrologic response model are able to provide understanding of the discharge behaviour from the catchment. Assessing the soil moisture state through such simulations may then help in assessing flood risk.

## 5   Conclusions

Simulations with the spatially explicit Alpine3D model were performed for the area of Davos. The recent update of the soil module of SNOWPACK, which provides the surface scheme for the Alpine3D model, shows satisfactory results for simulating soil moisture at 7 stations with soil moisture measurements in the area around Davos. The comparison included measurements at 10, 30 and 50 cm depths, and at 4 stations also at 80 and 120 cm depths. Correlation coefficients show that generally, the temporal variability is adequately captured. However, often a bias between simulated and measured soil moisture was found, as well as between two sensors at the same site and the same depth.

In winter, the amount of soil freezing was higher in the Alpine3D simulations than indicated by the measurements. The soil moisture measurements also provide some clear indications of fluctuations in groundwater level above 120 cm depth. Ground water dynamics is not taken into account in the model, as the water table was fixed to the lower boundary of the soil column in the model domain. Also uncertainties in soil properties and measurements likely play an important role in discrepancies between simulations and measurements.

Relating the water flux at 30 cm depth in the soil to streamflow in the Dischma catchment using a travel time distribution approach provided a higher agreement with observed streamflow than directly using the water flux at the top of the soil or at 60 cm depth. This may be a result of the (on average) relatively shallow layer of soil, which influence the near-surface water dynamics in Alpine terrain and is important to consider in the simulations. The analysis of events with high rainfall or snowpack runoff with return periods of approximately 15 and 30 times per year, respectively, showed that event and peak runoff coefficients using measured discharge were found to correlate with the simulated soil moisture state at the onset of the

events. Runoff coefficients for both the event as well as the peak were higher when the soil saturation was higher and vice versa. This effect was found to be stronger for snowpack runoff than for rainfall. Also runoff coefficients using simulated discharge exhibited a stronger relationship with initial soil saturation. The fact that the simulated soil moisture state could be related to the effect on measured streamflow, indicates that soil module of the SNOWPACK model in the Alpine3D model framework

can successfully assess the predisposition of the catchment for flood risk assessments.

**Acknowledgements**

This research has been conducted in the framework of the IRKIS project supported by the Office for Forests and Natural Hazards of the Swiss Canton of Grisons (Dr. Chr. Wilhelm), the region of South Tyrol (Italy) and the community of Davos. Contributions from the Swiss National Science Foundation (SNF: 200021_150146 and 200021E-160667) are further acknowl-

edged. We also thank Tobias Jonas for installing and maintaining the soil moisture measurements for stations Uf den Chaiseren, Grossalp and SLF2, and S. Valär for kindly allowing the installation of soil moisture sensors and weather stations on his property. The Alpine3D, StreamFlow and SNOWPACK models as well as the MeteoIO preprocessing library are available under a LGPLv3 license at http://models.slf.ch. The soil moisture measurements, in-situ meteorological measurements from the SensorScope stations, as well as interpolated meteorological model driving data at the soil measurement sites (enabling off-line

in-situ simulations), are available at EnviDat (doi: 10.16904/17).

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

**Table 1.** Yearly, winter months (DJF) and summer months (JJA) precipitation sums from heated rain gauges in the area around Davos. In brackets the percentage that falls as snow, based on measured air temperature below $1.2°C$ calculated on half-hourly measurements. The last line lists the average over the 10 year period 2005-2014.

| Year | Precipitation year mm (% snow) | Precipitation DJF mm (% snow) | Precipitation JJA mm (% snow) | Precipitation year mm (% snow) | Precipitation DJF mm (% snow) | Precipitation JJA mm (% snow) |
|---|---|---|---|---|---|---|
| | Davos (1590 m) | | | Weissfluhjoch (2540 m) | | |
| 2011 | 1062 (21%) | 145 (77%) | 409 (0%) | 1368 (47%) | 184 (95%) | 491 (7%) |
| 2012 | 1633 (42%) | 717 (83%) | 516 (0%) | 2337 (63%) | 1096 (100%) | 722 (6%) |
| 2013 | 1085 (28%) | 261 (82%) | 277 (0%) | 1590 (48%) | 400 (98%) | 476 (11%) |
| 2005-2014 | 1168 (28%) | 302 (76%) | 453 (0%) | 1659 (52%) | 440 (97%) | 648 (7%) |

**Table 2.** List of stations and measured quantities at the stations that are used in this study. (X): measured and used in this study, (-): not measured, (u): unventilated (temperature) or unheated (rain gauge), (v): ventilated, (h): heated rain gauge. VWC shallow denotes soil moisture sensors at 10, 30 and 50 cm depth, VWC deep denotes soil moisture sensors at 80 and 120 cm depth.

| Station name | Type | Elevation (m) | TA | RH | TSS | Wind speed | Snow depth | Rain gauge | ISWR | RSWR | ILWR | VWC shallow | VWC deep |
|---|---|---|---|---|---|---|---|---|---|---|---|---|---|
| Bärentalli | IMIS | 2560 | u | u | X | X | X | u | - | X | X | - | - |
| Flüelapass | IMIS | 2390 | u | u | X | X | X | u | - | X | X | - | - |
| Frauentobel | IMIS | 2330 | u | u | X | X | X | u | - | X | X | - | - |
| Gatschiefer | IMIS | 2310 | u | u | X | X | X | u | - | X | X | - | - |
| Grüniberg | IMIS | 2300 | u | u | X | X | X | u | - | X | X | - | - |
| Madrisa | IMIS | 2140 | u | u | X | X | X | - | - | X | X | - | - |
| SLF | IMIS | 1560 | u | u | X | X | X | - | - | X | X | X | X |
| Grossalp | IRKIS | 1960 | v | v | X | X | X | u | - | X | X | X | X |
| Uf den Chaiseren | IRKIS | 1590 | v | v | X | X | X | u | - | X | X | X | X |
| Dorfji | SENS[1] | 1813 | - | - | - | - | - | - | - | - | - | X | - |
| Golf Course | SENS[1] | 1537 | - | - | - | - | - | - | - | - | - | X | X |
| Pischa | SENS[1] | 2156 | - | - | - | - | - | - | - | - | - | X | - |
| Stillberg | SENS[1] | 2218 | - | - | - | - | - | - | - | - | - | X | - |
| Davos | SMN[2] | 1596 | - | - | - | - | - | h | X | - | X | - | - |
| Weissfluhjoch | COMBI[3] | 2536 | v | v | X | X | X | h | X | X | X | - | - |

[1] SENS: Sensorscope station.

[2] SMN: SwissMetNet station (MeteoSwiss).

[3] COMBI: Combination of IMIS, SwissMetNet and other instrumentation.

**Table 3.** Land use classes and corresponding soil initialisations.

| Land use class | Area (%) | Soil 0-60 cm | Soil 60-300 cm |
|---|---|---|---|
| Rock | 29.2 | loamy sand | loamy sand |
| Alpine meadow | 21.1 | silt loam | sandy loam |
| Rough pasture | 15.5 | silt loam | sandy loam |
| Mixed forest | 12.9 | silt loam | sandy loam |
| Bush | 7.3 | silt loam | sandy loam |
| Bare soil | 6.0 | silt loam | sandy loam |
| Glacier, ice, firn | 3.2 | ice | ice |
| Pasture | 2.6 | silt loam | sandy loam |
| Water | 1.0 | water | water |
| Settlements | 0.8 | rock | rock |
| Road | 0.5 | rock | rock |
| Wetland | 0.1 | silt loam | sandy loam |
| Vegetables | <0.1 | silt loam | sandy loam |

**Table 4.** List of parameters for the soil types for saturated water content ($\theta_\mathrm{s}$), residual water content ($\theta_\mathrm{r}$), the van Genuchten parameters $\alpha$ and $n$, the saturated hydraulic conductivity ($K_\mathrm{sat}$ [1]), the density of soil particles ($\rho_\mathrm{p}$), the thermal conductivity of soil particles ($\lambda$) and the specific heat of soil particles ($c_\mathrm{p}$).

| Name | $\theta_\mathrm{s}$ [1] $(\mathrm{m^3\,m^{-3}})$ | $\theta_\mathrm{r}$ [1] $(\mathrm{m^3\,m^{-3}})$ | $\alpha$ [1] $(\mathrm{m^{-1}})$ | $n$ [1] $(\text{-})$ | $K_\mathrm{sat}$ [1] $(\mathrm{m\,s^{-1}})$ | $\rho_\mathrm{p}$ $(\mathrm{kg\,m^{-3}})$ | $\lambda$ $\mathrm{W\,m^{-1}\,s^{-1}}$ | $c_\mathrm{p}$ $\mathrm{J\,kg^{-1}\,K^{-1}}$ |
|---|---|---|---|---|---|---|---|---|
| Loamy sand | 0.390 | 0.049 | 3.475 | 1.746 | $1.22 \cdot 10^{-5}$ | 2600 [2] | 0.9 [2] | 1000 [2] |
| Sandy loam | 0.387 | 0.039 | 2.667 | 1.449 | $4.43 \cdot 10^{-6}$ | 2600 [3] | 2.5 [3] | 801 [3] |
| Silt loam | 0.439 | 0.065 | 0.506 | 1.663 | $2.11 \cdot 10^{-6}$ | 2700 [3] | 2.5 [3] | 871 [3] |

[1] ROSETTA class average parameters (*Schaap et al.*, 2001).

[2] *Bachmann et al.* (2001).

[3] *Ochsner et al.* (2001).

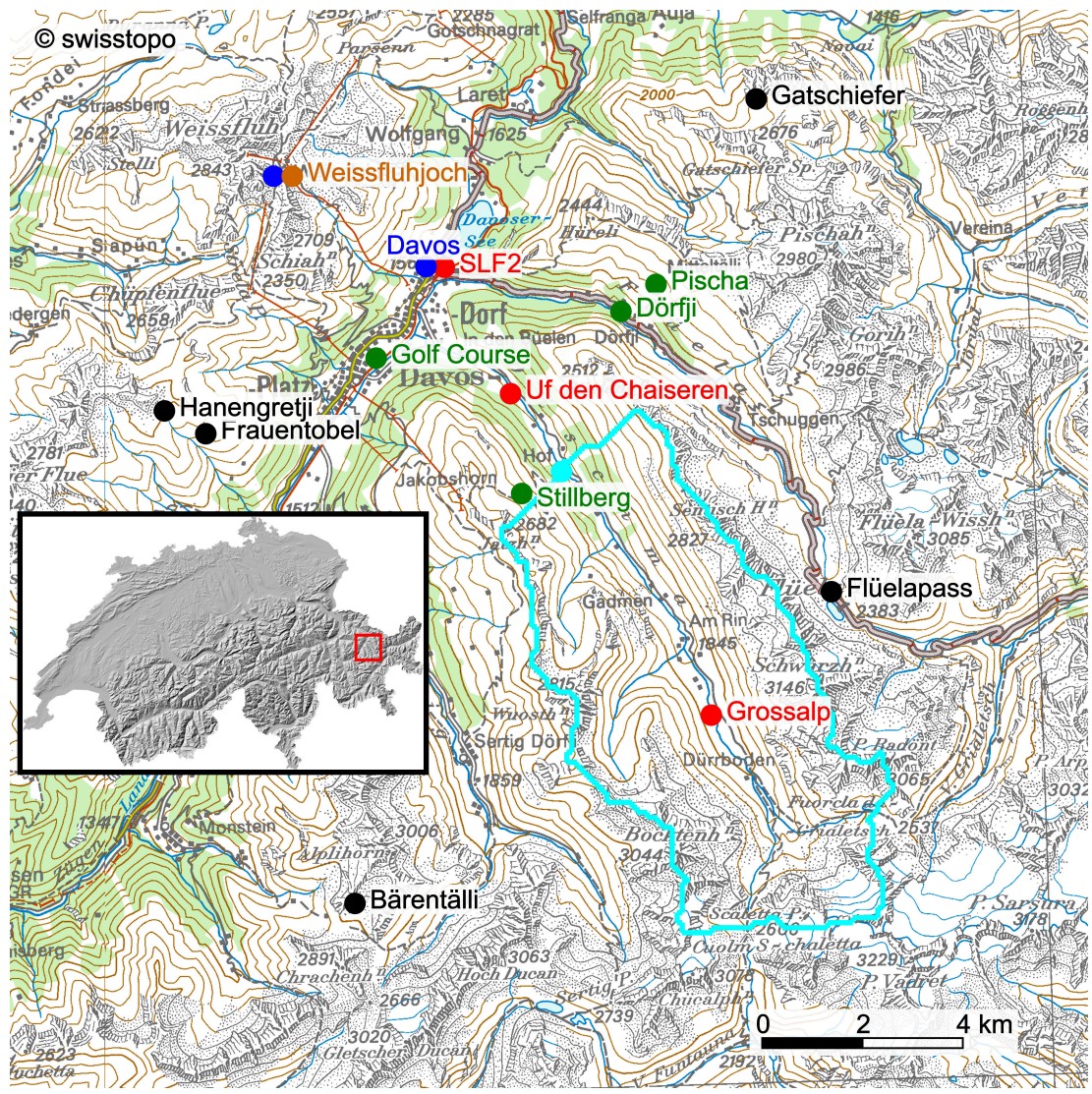

**Figure 1.** Topographical map of the simulated domain, showing the locations of the stations. IMIS stations are shown in black, IRKIS stations in red, SensorScope stations in green, SwissMetNet stations in blue and Weissfluhjoch in brown. The Dischma catchment and the gauging station measuring streamflow in the Dischmabach at the outlet of the Dischma catchment are shown in cyan. The inset shows the location of the simulation domain (red square) in Switzerland. Maps reproduced by permission of swisstopo (JA100118).

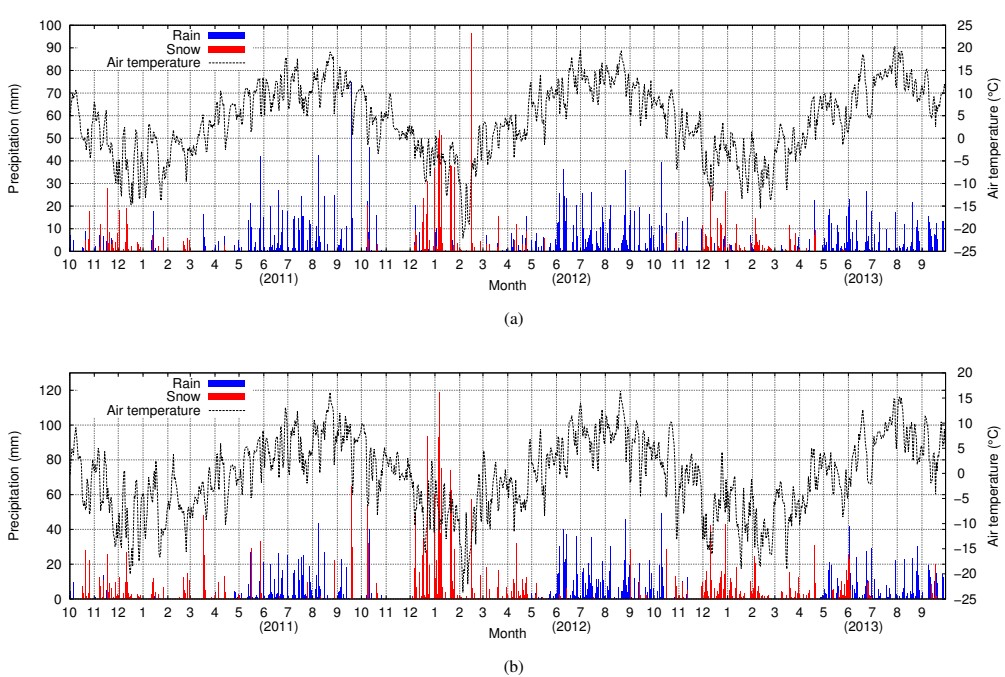

**Figure 2.** Daily rain and snowfall amounts and daily average air temperature for Davos, 1590 m (a) and Weissfluhjoch, 2536 m a.s.l. (b). The separation of precipitation in rain and snowfall is done for half-hourly measurements, using an air temperature threshold of 1.2 °C.

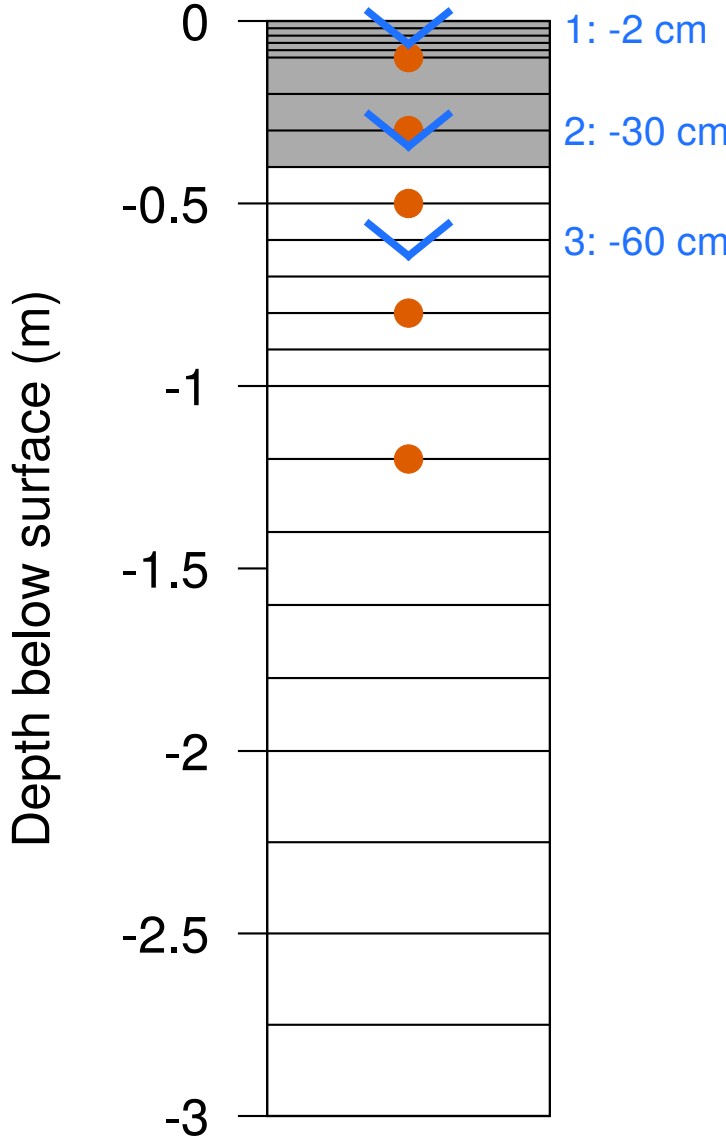

**Figure 3.** Soil layering as used in the Alpine3D model. The three water fluxes used to drive the hydrologic response model are shown in blue arrows. The soil moisture measurements are indicated by brown circles. The grey area is denoting the part of the soil where the initial soil saturation at the onset of rainfall or snowmelt events was determined.

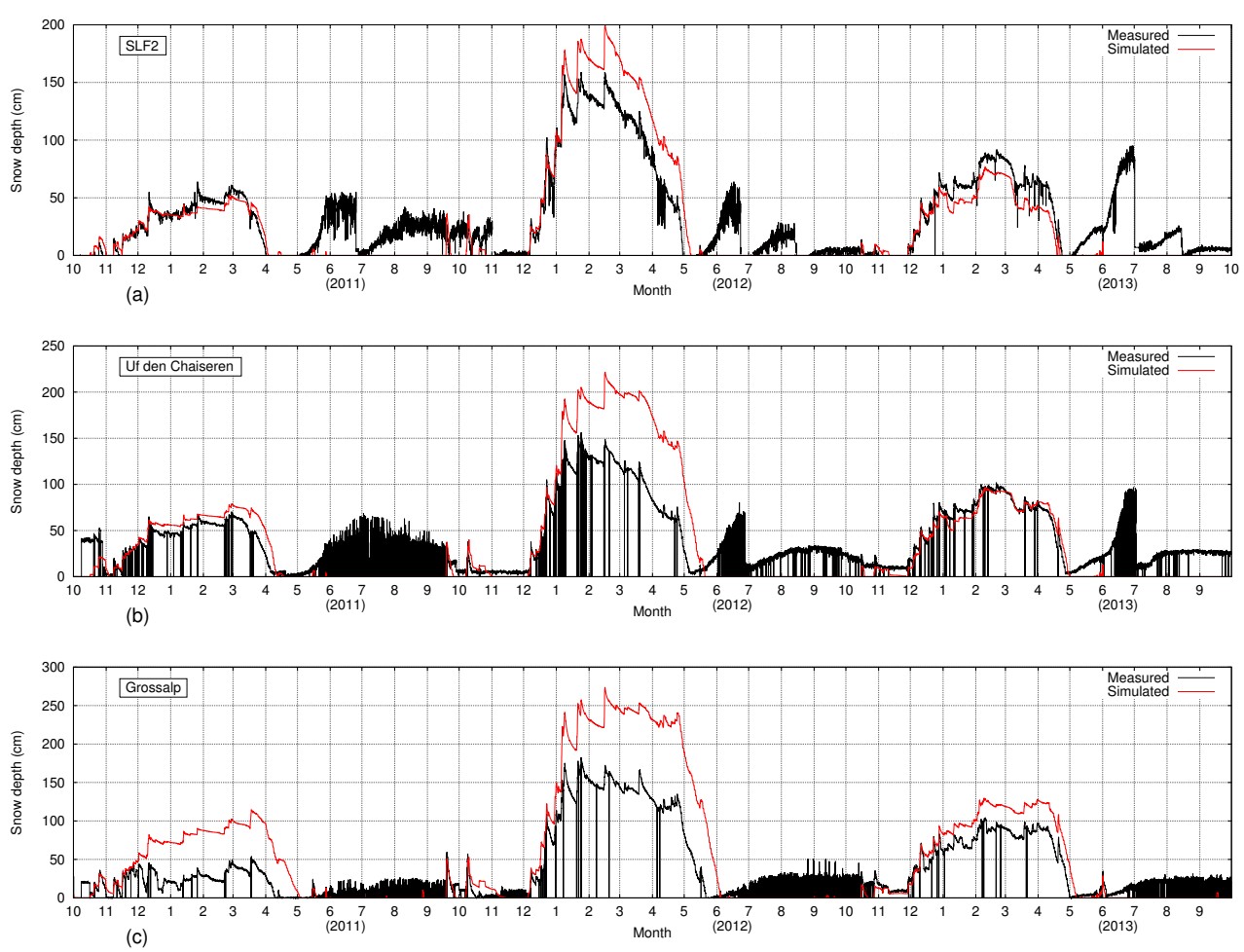

**Figure 4.** Measured and simulated snow depth for stations SLF2 (a), Uf den Chaiseren (b) and Grossalp (c) for the period October 2010 to October 2013. Noisy signals in the summer months arise from grass growth below the sensor.

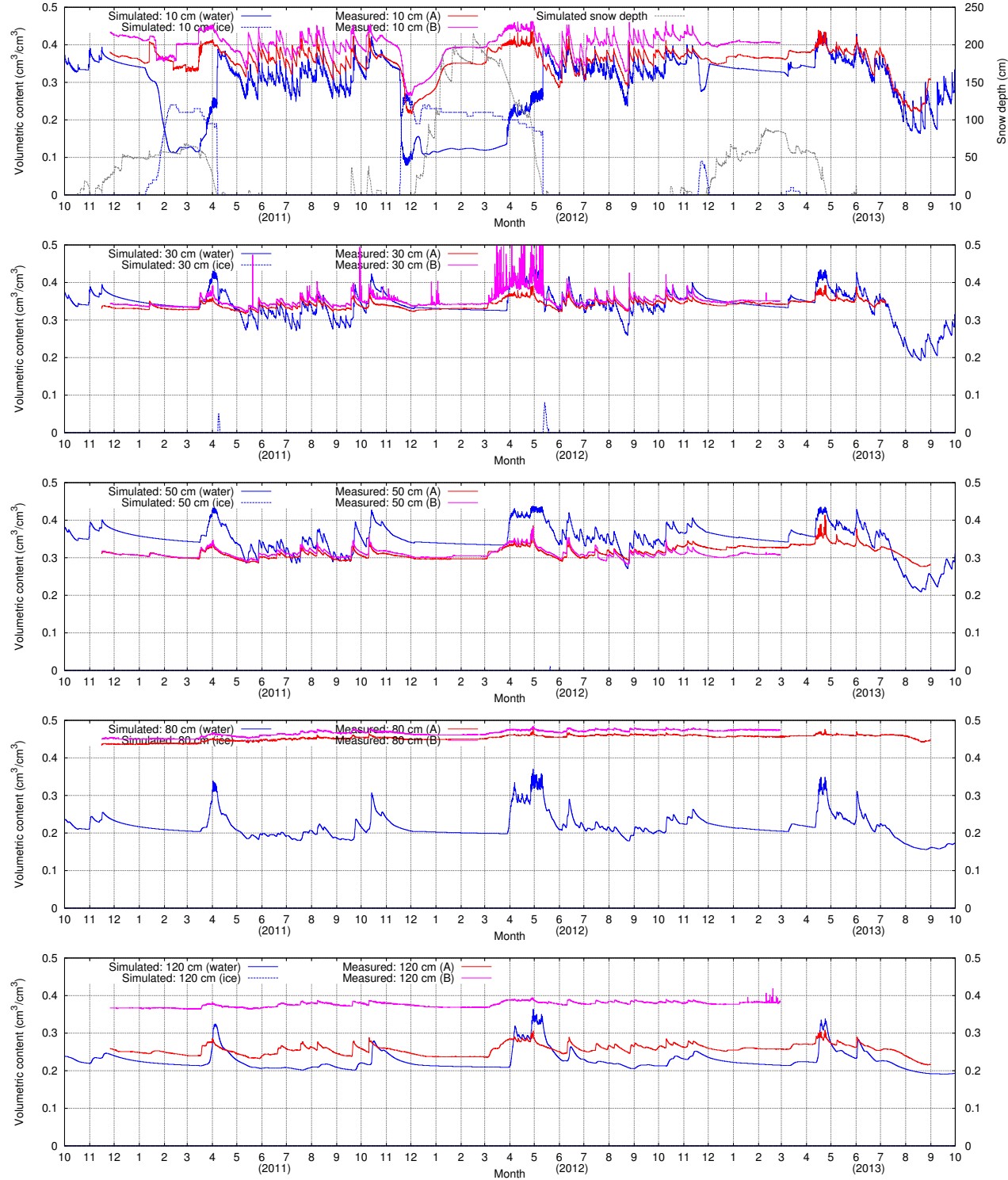

**Figure 5.** Measured and simulated soil moisture at the IRKIS station SLF2, for (from top to bottom) 10, 30, 50, 80 and 120 cm depth for the period October 2010 to October 2013. In the upper panel, also simulated snow depth is shown.

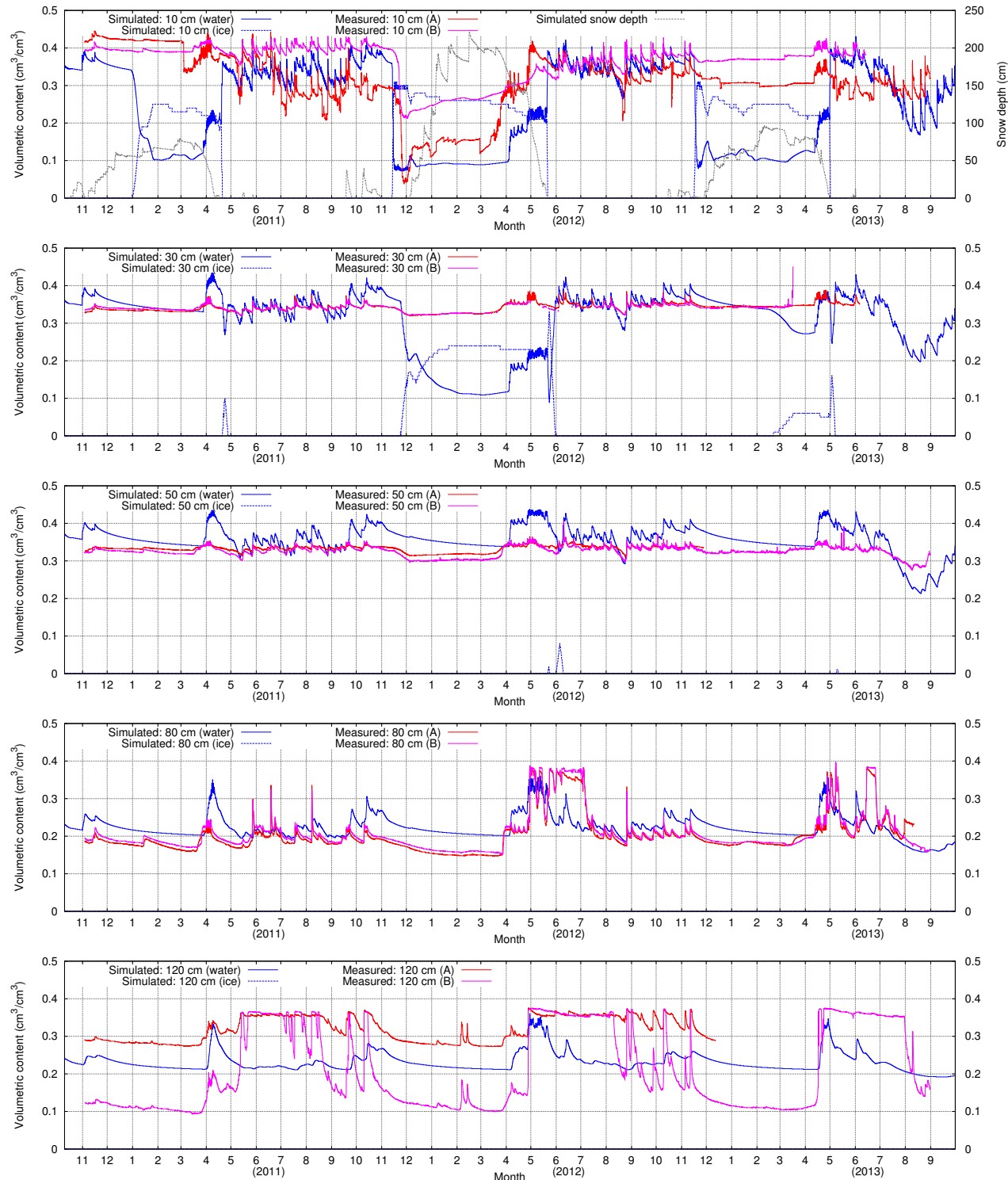

**Figure 6.** Measured and simulated soil moisture at the IRKIS station Uf den Chaiseren, for (from top to bottom) 10, 30, 50, 80 and 120 cm depth for the period October 2010 to October 2013. In the upper panel, also simulated snow depth is shown.

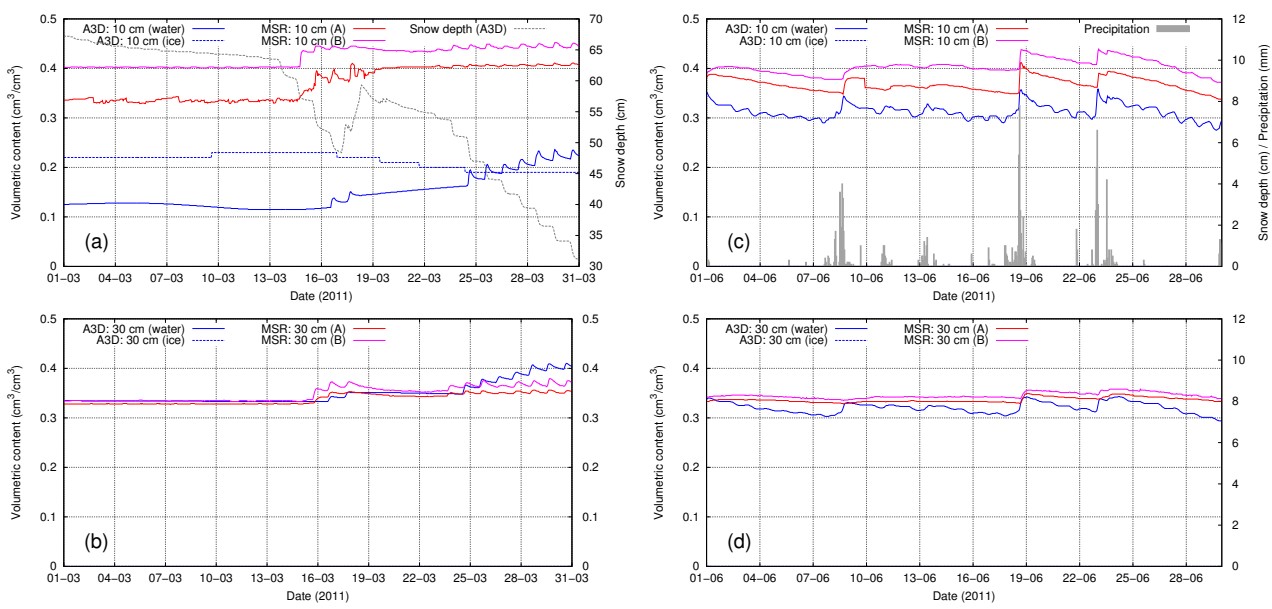

**Figure 7.** Measured (MSR) and simulated (A3D) soil moisture at the IRKIS station SLF2, for 10 cm depth (a, c) and 30 cm depth (b, d), during the snow melt season (a, b) and a snow-free summer month (c, d). In (a) simulated snow depth and in (c) precipitation is shown.

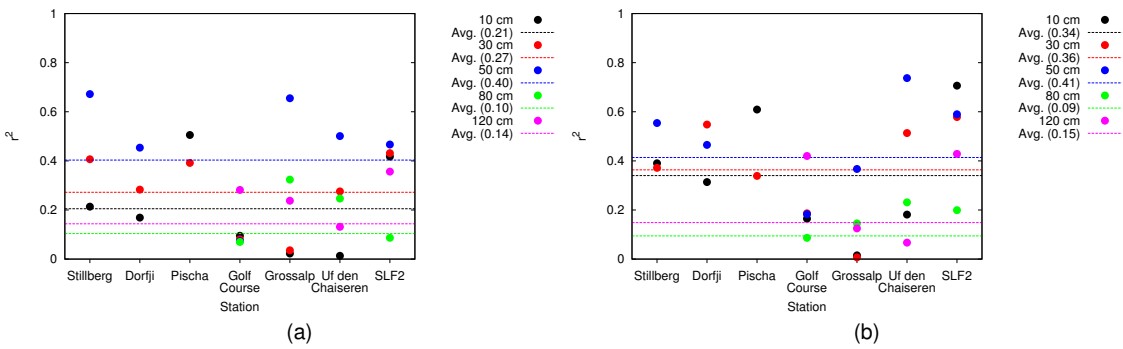

**Figure 8.** $r^2$ between measured and simulated soil moisture for the full period (a) and the summer months (b) for the 7 soil moisture stations. Dashed lines indicate the average value determined over all stations.

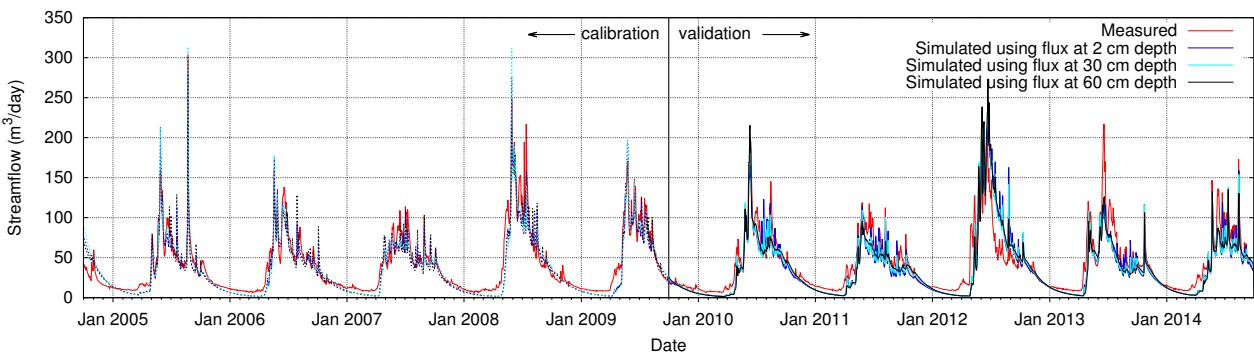

**Figure 9.** Measured and simulated daily streamflow for the outlet of the Dischmabach. Dashed lines denote the calibration period, solid lines denote the validation period. Major ticks on the x-axis are drawn at January 1 of each year, minor ticks are drawn at every other first of the month.

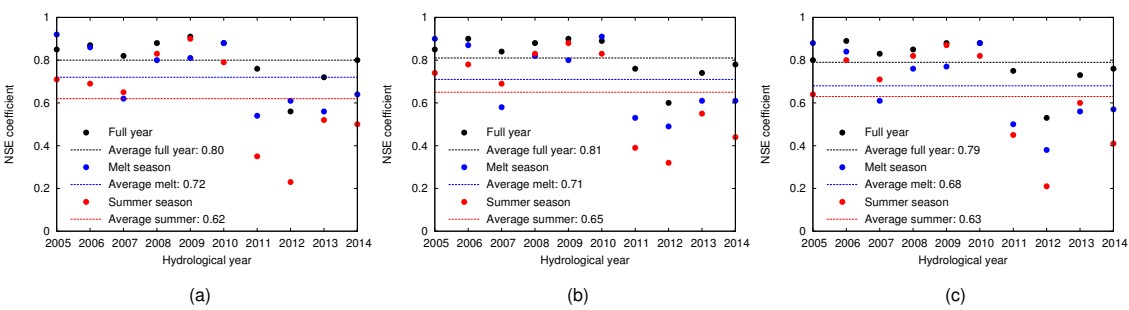

**Figure 10.** NSE coefficients for simulated daily streamflow for the outlet of the Dischmabach, using the 2 cm (a), 30 cm (b) or 60 cm (c) water flux in the soil layers.

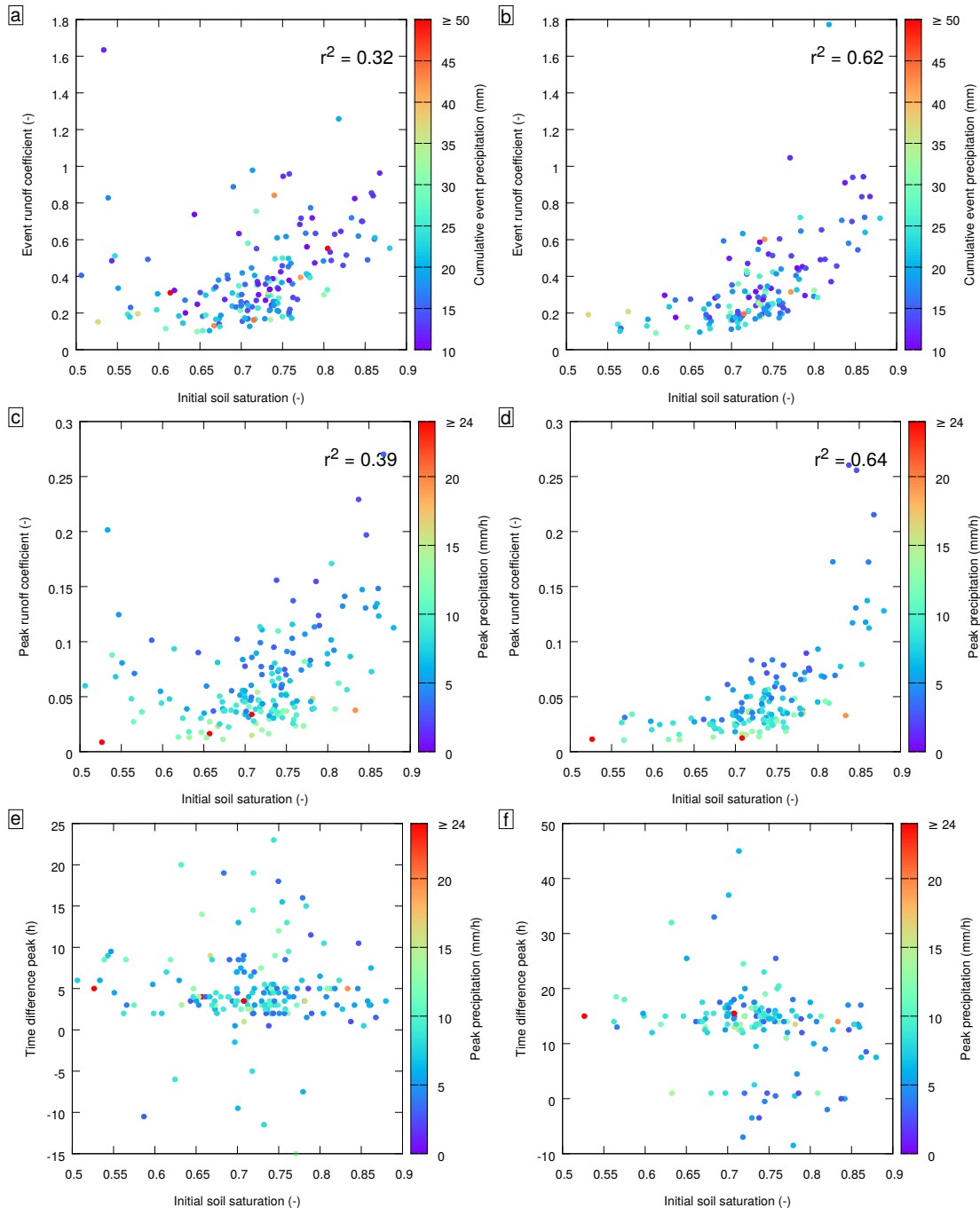

**Figure 11.** Rainfall event runoff coefficients for measured discharge as a function of initial soil saturation in the upper 40 cm of the soil (a) and similar for simulated discharge (b). Peak rainfall runoff coefficients for measured discharge as a function of soil saturation (c) and similar for simulated discharge (d). Time difference between peak rainfall and measured peak discharge (e) and similar for simulated peak discharge. Points are coloured according to the event rainfall sum (a and b) or the peak rainfall (c, d, e and f).

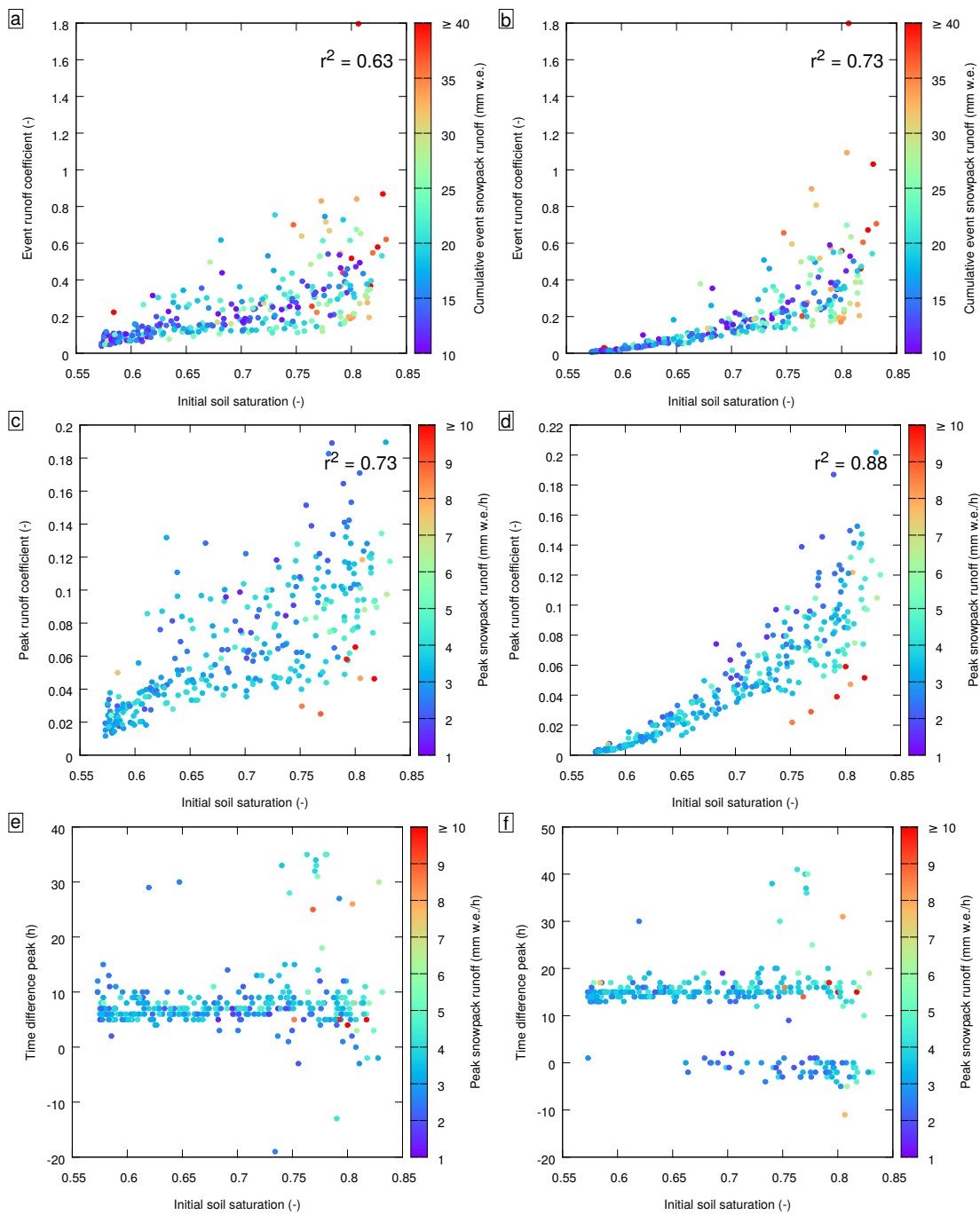

**Figure 12.** Snowpack runoff event runoff coefficients for measured discharge as a function of initial soil saturation in the upper 40 cm of the soil (a) and similar for simulated discharge (b). Peak snowpack runoff runoff coefficients for measured discharge as a function of soil saturation (c) and similar for simulated discharge (d). Time difference between peak snowpack runoff and measured peak discharge (e) and similar for simulated peak discharge. Points are coloured according to the event snowpack runoff sum (a and b) or the peak snowpack runoff (c, d, e and f).