# Peer review of "Simulating the influence of snow surface processes on soil moisture dynamics and streamflow generation in an Alpine catchment"

_Hydrology and Earth System Sciences, 2016_

## Referee Comment (RC1) · Anonymous Referee #1 · 17 Mar 2017

In the manuscript "Influence of snow surface processes on soil moisture dynamics and streamflow generation in alpine catchments", the authors present a comprehensive modeling study using the model Alpine3D which was complemented with new descriptions for simulating soil moisture and streamflow. The distributed model was forced using meteorological station data at several points and validated by means of snow depth, soil moisture, and runoff measurements.

General Comments

The manuscript presents a modeling study using a detailed set of modules and methods to tackle the challenge of simulating the hydrology in a complex mountainous catchment with a fully distributed, spatially highly resolved model. The focus lies on

the simulation of snow depth, soil moisture and the respective interplay of precipitation, snow melt, and runoff dynamics. The study gives valuable insights in the involved hydrological processes. The manuscript is well elaborated and written and is technically of very high quality. I recommend its publication after minor revisions. Generally, the presented analysis is a bit incomplete because of the lack of a groundwater description in the model. This is mentioned in the manuscript at the respective sections. But it should be emphasized even more that this is a major shortcoming of the study and it should be addressed in future work with the model setup. Another criticism is the description of the presented streamflow model. It is not quite clear to me how it was coupled to the model and what the flux input at or from different depths mean (sections 3.2 and 4.3). As I understand it, the water fluxes from the soil model at three different depths (lateral flux or excess out of the respective soil layer?) were taken and "streamed" into an external streamflow model. This streamflow model is calibrated using the respective fluxes which produces the shown streamflow simulations for three different depths. This approach is quite unusual and definitely needs further explanation in the manuscript. Why is the flux taken separately from the depths and not combined? The runoff dynamics clearly reveal that a groundwater module is missing. But this missing groundwater module could be "replaced" by a calibrated low flow component of the water flux (baseflow) which seems to be totally missing (Fig. 7, underestimated low flow / baseflow in the winter months). All other presented findings regarding soil moisture, freezing, as well as event-based precipitation and melt are well elaborated and very interesting. Some more questions that need clarification are listed in the following specific comments.

Specific Comments

P.1 L. 6: "in close proximity to" instead of "in close proximity of"

P. 1 L. 9-15: "Streamflow simulations performed with a spatially-explicit hydrological model using a travel time distribution approach coupled to Alpine3D provided a closer agreement with observed streamflow at the outlet of the Dischma catchment when

including 30 cm of soil layers. Performance decreased when including 2 cm or 60 cm of soil layers. This demonstrates that the role of soil moisture is important to take into account when understanding the relationship between both snowpack runoff and rainfall and catchment discharge in high alpine terrain. " The differences in NSE for three simulations are so small that I would not give this strong statement. It is also not at all an evidence for your second statement as you show no simulations without the new soil model, see comment below (P. 10 L. 18/19). It is for sure correct that soil moisture has to be taken into account but you show no real proof in this work.

P.1 L. 17: "which shows" instead of "and this shows"

P. 3 L. 7: Rephrase: "The measurement site Weissfluhjoch (WFJ), which is focused on snow-related measurements, as well as several permanent meteorological stations are located in close proximity to the area." instead of "The measurement site Weissfluhjoch (WFJ), which is focussed on snow-related measurements, is located in close proximity of the area, as well as several permanent meteorological stations."

P. 3 L. 14: "of total precipitation" instead of "of all precipitation"

P. 4 L. 7: Better use "focused", not "focussed" (see also above P. 3 L. 7)

P. 5 L. 14: Lower computational costs compared to what other approach? Please add an example for clarification!

P. 5 L. 21: Remove brackets in citation!

P. 6 L. 13: Rephrase the first two sentences / the beginning of this section ("Two important components to initialise Alpine3D simulations are the digital elevation model (DEM) for the Davos area, provided by the Swiss Federal Office of Topography (swisstopo). Also the soil has to be initialised for each pixel, although limited information is available.") e.g.: "Two important components to initialise Alpine3D simulations are the digital elevation model (DEM) and distributed soil information. The DEM is provided. . . "

[Figure]

P. 7 L. 6: Either remove "on a computer cluster from 2008." or preferably provide some more information about the HPC system (e.g. type and clock speed of nodes). I guess the 14 hours per year using 36 CPU cores are the necessary wall clock time (or CPUh?). Please add this information in the manuscript.

P. 7 L. 11: Remove "also".

P. 7 L. 12: Why didn't you additionally inspect hourly values if you have the respective measurements? You could add at least some examples for showing the model performance on a smaller, hourly timescale, which would be very interesting to see.

P. 8 L. 6 ff and above and Figures 3 – 5: Consequently use one throughout the manuscript: either "snow depth" or "snow height" (personally, I prefer "depth").

P. 8 L. 12 ff and Fig. 3: Please try to remove the measurement errors in Fig. 3 (high frequency fluctuations, especially in the summer months)! In June / July 2012 and 2013, the model seems to miss the measured spring snow fall at stations (a) and (b). Why does this happen? Add a respective explanation in the manuscript.

P. 8 L. 15: To be consistent with the section title of 4.1 either remove "Measurements and Simulations" or add it in 4.1

P. 9 L. 20: "S1" instead of "S3"

P. 9 L. 27: Are the r2 values calculated using the daily or hourly values? I guess daily, but please add this in the manuscript for clarification.

P. 10 L. 12: Remove "us".

P. 10 L. 15: Please either explain your concept of the "virtual lysimeter" or use another notion! I think you are referring to the water fluxes at the three depths, but this is not clear here.

P. 10 L. 18/19: I am not sure if I understand this right, but the statement "The results suggests that the updated soil module of SNOWPACK is contributing to a better

prediction of streamflow in the summer months. " is misleading or drawn without any evidence. You show no Alpine3D runoff result without the new model, because – as I understand it – there was no soil moisture or runoff description in the model before. So a valid statement would be something like "The results show that the new soil module of SNOWPACK is enabling a simulation of streamflow."

P. 11 last paragraph of section 4.4: The conclusions here are of course valid but were somehow clear before your study and should be underlined with existing literature.

Fig. 2, 3, 4, 5 and S1 – S5: Please add the year to the time-axis. This makes it much easier to look at when you write about single years in the text.

Fig. 7, caption: typo "tics"

Fig. 8, caption: I cannot see any data points plotted on the x-axis as stated in the caption. When you add them, please add the real value somehow because it is of interest how negative the NSE values are in these periods.

---

## Referee Comment (RC2) · Anonymous Referee #2 · 19 May 2017

The manuscript mainly presents a comprehensive, multi-objective model validation study of an extension of the complex, physics-based Alpine 3D model. The model's performance to represent streamflow, snow depth, soil moisture and soil temperature/freezing is analyzed for 3 years in a Swiss mountain catchment and its surrounding. This evaluation is a framed by the context of the importance of soil moisture as a pre-disposition for floods. This framing – for my taste – is a bit wanted, and the title is somehow misleading. However, the quality of the model validation study is of a high standard and certainly of high importance. Especially the parallel evaluation of snow, soil and streamflow representation is cool. Although the overall impression of the article is very good, I do have some moderate remarks. The manuscript is certainly within

the scope of HESS, it my knowledge of original content and can be a valuable article after my concerns are addressed.

General comments:

- As mentioned before, I find the title a bit misleading, as it reads as the influence of snow processes on soil moisture and streamflow is analyzed. Instead, the focus is clearly on the model validation to reproduce the linkage between snow, soil and streamflow. I would recommend a rephrasing of the title.

- Alike reviewer 1 (I haven't read his comments until I finished my review), I do not understand the usage of the soil water fluxes for streamflow generation. As reviewer 1 wrote to the point, why should one just take one of the fluxes (-2cm indicating surface runoff, -30 cm indicating interflow, -60 cm indicating baseflow). I also had a look in the cited publication, yet I find the entire concept very unusual and irritating. Because of this (I guess), the interpretation of the influence of soil moisture (Page 10, lines 14 ff) on streamflow is a bit simple. E.g. "neglecting the soil layers almost completely, by routing the 2 cm flux to the runoff model, is reducing the model efficiency". – This is logical as you neglect interflow and baseflow in summer months and interflow widely considered to be the dominant process in alpine catchments. Hence, the concept of this streamflow generation needs to be clarified and its strong limitation in terms of dynamic runoff generation should be discussed. Furthermore, effects of this simplified approach on the interpretation should be discussed.

- The description of the different soil layers is unclear: You introduced increasing soil layer magnitudes (from 2cm to 40 cm) up to a soils depth of 300 cm in the model. However, you take water fluxes from 2, 30 and 60 cm. Moreover you compare these to soil moisture measured at 10, 30, 50, 80, and 120 cm depth. And finally, you take the average of the upper 40 cm (page 10, line 25) as the soil moisture state within the catchment. As these number do not match at a first glance, a clarification is advisable. Maybe a sketch would help.

[Figure]

- In the manuscript, the SNOWPACK and Alpine3D are described as two separate models (e.g. in the model description and partly in the introduction). But as written in the Conclusion, SNOWPACK is a module of Alpine3D and as I understand an integrated part of Alpine3D. This should be clarified throughout the text, especially in the beginning (Aims section)

- As the Dischma catchment is an alpine catchment I assume that skeleton fraction is a major issue, both for measuring the "correct" soil moisture as well as for simulating the soil moist dynamics. Please, clarify how and if the skeleton fraction was considered in the pedo-transfer-function and how it was considered in the selection of the measuring location (and how representative the selection in terms of skeleton fraction is). Moreover, please discuss if the found biases in the soil moisture and soil temperature simulations can be explained by skeleton fraction. Finally (I hope I did not miss it), how do the soil types of all measuring stations represent the soil types in the Dischma catchment.

- The description of the meteorological data is quite long and very detailed. I would suggest to just briefly describe the table 2.

Specific comments:

- Page 1, line 11, and 12: Please clarify the word "including", as you do not combine the three layers.

- Page 2, line 29: "small scale surface processes". Please, specify the scale.

- Page 3 ,line 5: Please, specify the catchment size

- Page 3, line 13 ff & Figure 2: How did you separated snow from rain here.

- Page 3, and Table 1: A comparison to the long term norm period would be interesting

- Page 4 and Figure 1: "Golfplatz" in the Figure versus "Golf course" in the text. "SLF2" site is named "SLF" in the map. How were the boarders of the Dischma catchment

defined (topography based from the model?). I would recommend some light, partly transparent background color for the names, to improve readability. I have to admit, I am not a fan of topographic maps as background, especially if the legend is missing. Any chance to replace it with a more generalized map?

- Page 6, line 5 ff: Are the interpolations done for each time step?

- Page 6, line 14: I do not think that "initialization" is the correct term. Is it not parameterization?

- Page 7, line 21. "sub-catchments" – so is this approach some kind of HRU approach?

- Page 7, line 33: Again, the soil moisture is calculated for the first 40 cm. Can you clarify its relation to the 30 cm stated before and after.

- The definition of a rainfall event is a bit broad. Do you used mowing 12 h sum? What if a rainfall event is ended by falling below the 3mm thresholds criteria, but followed by a >10mm event again. Why do you choose a time window of 12 mm. Did you do any concentration time analysis?

- Page 8, line 14, and Figure 3. A comment on the vegetation growth (?) during summer would be nice.

- Page 9, line 27 ff. In my opinion, the r2 is not the appropriate statistical measure here, as it does not consider any systematic offsets/biases. The application of the RMSE or similar would be more fair. Furthermore, can you set your results in light of other models of soil moistures in alpine terrains? Also to show that your results are pretty good.

- Page, line 10: "however, ...." Isn't this finding clear and logical as you only consider "deeper" water fluxes

I am looking forward to the revised manuscript.

---

## Author Comment (AC2) · 13 Jun 2017

**1 Reviewer 2**

The manuscript mainly presents a comprehensive, multi-objective model validation study of an extension of the complex, physics-based Alpine 3D model. The model's performance to represent streamflow, snow depth, soil moisture and soil temperature/freezing is analyzed for 3 years in a Swiss mountain catchment and its surrounding. This evaluation is a framed by the context of the importance of soil moisture as a pre-disposition for floods. This framing – for my taste – is a bit wanted, and the title is somehow misleading. However, the quality of the model validation study is of a high standard and certainly of high importance. Especially the parallel evaluation of snow, soil and streamflow representation is cool. Although the overall impression of the article is very good, I do have some moderate remarks. The manuscript is certainly within the scope of HESS, it my knowledge of original content and can be a valuable article after my concerns are addressed.

*We thank the reviewer for the constructive comments and positive remarks about our study. We will take them into consideration when revising the manuscript. As we will discuss below, we agree that the title may be considered confusing and we propose a modification of the title. However, we would like to keep the framing of the study of soil moisture as a predisposition for flooding. This notion is introduced in the introduction section, with, in our opinion, appropriate citations. Furthermore, we do find evidence that our model framework is able to reproduce the relationship between soil saturation at the onset of large rainfall or snowmelt events and the discharge behaviour in the Dischma catchment.*

**1.1 General comments:**

- As mentioned before, I find the title a bit misleading, as it reads as the influence of snow processes on soil moisture and streamflow is analyzed. Instead, the focus is clearly on the model validation to reproduce the linkage between snow, soil and streamflow. I would recommend a rephrasing of the title.
  *We agree that the title is somewhat misleading, so we suggest to add the word "Simulating" in the beginning of the title, so we propose now: "Simulating the influence of snow surface processes on soil moisture dynamics and streamflow generation in alpine catchments".*

- Alike reviewer 1 (I haven't read his comments until I finished my review), I do not understand the usage of the soil water fluxes for streamflow generation. As reviewer 1 wrote to the point, why should one just take one of the fluxes (-2cm indicating surface runoff, -30 cm indicating interflow, -60 cm indicating baseflow). I also had a look in the cited publication, yet I find the entire concept very unusual and irritating. Because of this (I guess), the interpretation of the influence of soil moisture (Page 10, lines 14 ff) on streamflow is a bit simple. E.g. "neglecting the soil layers almost completely, by routing the 2 cm flux to the runoff model, is reducing the model efficiency". – This is logical as you neglect interflow and baseflow in summer months and interflow widely considered to be the dominant process in alpine catchments. Hence, the concept of this streamflow generation needs to be clarified and its strong limitation in terms of dynamic runoff generation should be discussed. Furthermore, effects of this simplified approach on the interpretation should be discussed.
  *This comment is similar to the general comment of Reviewer 2 and for simplicity, we give the same response here: As both reviewers raise similar concerns, it is clear that we need to pay particular attention to describe the coupling between the Alpine3D and streamflow model in more detail and with more clarity when revising the manuscript. The streamflow model is a spatially explicit hydrologic response model at sub-catchment scale. Each sub-catchment is identified based on geomorphological analysis of the watershed. The model simulates the water storage dynamics in two soil compartments, namely an upper and lower one, of each sub-catchment using a travel time distribution approach. Outflow from the*

*upper compartment represents interflow, while that from the lower component represents baseflow. We would like to point out that our model is reproducing baseflow, albeit too low at the end of the winter. If one would be particularly interested in correctly representing baseflow, a recalibration of the streamflow model with a focus on the statistics for the winter period would allow to have a more accurate representation of baseflow. But we do not agree with both reviewers that the baseflow is absent. Furthermore, the streamflow model needs a surface scheme, to provide the influx into the system. For this, we use the Alpine3D model. However, it is somehow arbitrary where to draw the boundary between the surface scheme and the streamflow model. For this, we tested 3 scenarios: a soil flux at 2, 30 and 60 cm depth. So we do not use the fluxes combined, but we used the three fluxes as three different scenarios. Although it would be similar as running 3 separate simulations, with either 2, 30 or 60 cm of soil, this approach would have the disadvantage that specifying the lower boundary condition for the Alpine3D model becomes tricky. For example, at 3 m depth, one can assign a constant geothermal heat flux and a water table and this would hardly influence the snowpack dynamics. On the other hand, at 2 cm below the surface, a constant geothermal heat flux would provide a too strong heating of the snowpack, as the soil buffer is not represented. Therefore, we choose the approach of doing a single simulation, and extracting soil water fluxes at three depths in order to test how to achieve an optimal coupling between the surface scheme Alpine3D and the streamflow model. This is illustrated in Fig. 3 in Comola et al. (2015). We will rewrite the Methods section discussing the model coupling thoroughly, in order to better explain our approach.*

*Please find our response to other issues raised by the reviewer below.*

- The description of the different soil layers is unclear: You introduced increasing soil layer magnitudes (from 2cm to 40 cm) up to a soils depth of 300 cm in the model. However, you take water fluxes from 2, 30 and 60 cm. Moreover you compare these to soil moisture measured at 10, 30, 50, 80, and 120 cm depth. And finally, you take the average of the upper 40 cm (page 10, line 25) as the soil moisture state within the catchment. As these number do not match at a first glance, a clarification is advisable. Maybe a sketch would help.
  *We think this is a very good suggestion and we will add a descriptive illustration in the revised manuscript. Please find the new figure below as Fig. 2. Note that the layer spacing was erroneously reported as ranging from 2 cm to 40 cm, where it should have been from 2 cm to 25 cm. This will be corrected in the revised manuscript.*

- In the manuscript, the SNOWPACK and Alpine3D are described as two separate models (e.g. in the model description and partly in the introduction). But as written in the Conclusion, SNOWPACK is a module of Alpine3D and as I understand an integrated part of Alpine3D. This should be clarified throughout the text, especially in the beginning (Aims section)
  *The reviewer is correct, the SNOWPACK model serves as the snow and soil module for the distributed Alpine3D model. When revising the manuscript, we will pay attention that this is correctly formulated throughout the manuscript.*

- As the Dischma catchment is an alpine catchment I assume that skeleton fraction is a major issue, both for measuring the "correct" soil moisture as well as for simulating the soil moist dynamics. Please, clarify how and if the skeleton fraction was considered in the pedo-transfer-function and how it was considered in the selection of the measuring location (and how representative the selection in terms of skeleton fraction is). Moreover, please discuss if the found biases in the soil moisture and soil temperature simulations can be explained by skeleton fraction. Finally (I hope I did not miss it), how do the soil types of all measuring stations represent the soil types in the Dischma catchment.

*- We agree that the skeleton fraction in alpine catchment is an important factor to take into account. For example, the study by Rössler and Löffler (2010) demonstrates in a sensitivity study that changing the skeleton fraction has an impact on streamflow and soil moisture simulations. They particularly describe the effect of an increase in porosity, and associated changes in hydraulic parameters. However, the study by Rössler and Löffler (2010) also points out that spatial variability of the skeleton fraction is largely unknown. In the current version of the SNOWPACK model, which provides the surface scheme for the Alpine3D model, the skeleton fraction is not taken into account, as the SNOWPACK uses prescribed soil types. We actually found that for some sites we get an adequate soil moisture simulation without considering the skeleton fraction, whereas for other sites the simulations are showing less agreement with measurements. But these contrasting results indicate that with the current information, only ad-hoc modifications of the skeleton fraction are possible, as we cannot separate well enough between the soil moisture sites based on available information (land use and soil permeability). For further development of the SNOWPACK and Alpine3D model, this certainly is an area of attention. As discussed by Brakensiek and Rawls (1994), neglecting rock fragments in soil may overestimate hydraulic conductivity. Thus, the bias in soil moisture we found can be explained by an overestimation of hydraulic conductivity in the model, which would bring down liquid water faster. As wetter soils need more energy to freeze, the underestimation of soil moisture in the top layer may also result from this bias. The above mentioned points will be discussed in the revised manuscript.*

*- The representativeness of the soil moisture measurement sites is given by that the Grossalp and Pischa stations were located in the "alpine meadow" class, which is 21.1% of the land use coverage (see Table 4 in the original manuscript). The Uf den Chaiseren, Dorfji and Stillberg stations are located in the "mixed forest", "bush" and "bare soil" class, respectively, which is found in 12.9%, 7.3% and 6.0% of the Dischma catchment, respectively. The SLF2 and Golf Course stations would officially fall into the category of "settlement", but one would describe the area as "alpine meadow". We will add this information to the manuscript.*

- The description of the meteorological data is quite long and very detailed. I would suggest to just briefly describe the table 2.
  *When revising the manuscript, we will put effort in shortening this section.*

**1.2 Specific comments:**

- Page 1, line 11, and 12: Please clarify the word "including", as you do not combine the three layers.
  *As our description of the coupling to the streamflow model was clearly confusing in the original manuscript, we plan to revise this part of the abstract as: "Streamflow simulations performed with a spatially-explicit hydrological model using a travel time distribution approach coupled to Alpine3D provided a closer agreement with observed streamflow at the outlet of the Dischma catchment when driving the streamflow model with soil water fluxes at 30 cm depth. Performance decreased when using the 2 cm soil water flux, thereby mostly ignoring soil processes. This demonstrates that the role of soil moisture is important to take into account when understanding the relationship between both snowpack runoff and rainfall and catchment discharge in high alpine terrain. However, using the soil water flux at 60 cm depth to drive the streamflow model also decreased its performance, indicating that an optimal soil depth to include in the simulations exists."*

- Page 2, line 29: "small scale surface processes". Please, specify the scale. *This refers to 10-100m scale, on which wind drifts form, and for which local topography strongly influences the energy balance via the slope aspect, angle and local shading. We will amend the manuscript at this point.*

- Page 3 ,line 5: Please, specify the catchment size.
  *We will report that the catchment size that is represented by the gauging station is 43.3 $km^2$, as reported by the Swiss Federal Office of the Environment (Federal Office for the Environment (FOEN), 2017).*

- Page 3, line 13 ff & Figure 2: How did you separated snow from rain here.
  *In the manuscript, we did all separations of precipitation in rain and snowfall based on an air temperature threshold of 1.2 °C for half-hourly measurements. We will specify this in the manuscript where necessary.*

- Page 3, and Table 1: A comparison to the long term norm period would be interesting
  *We agree with this suggestion. We now add the 10-year averages to the table, which corresponds to the period for which the streamflow simulations were performed. Note that we came across an inconsistency. The data shown was not based on the same meteorological dataset as used for the Alpine3D simulations. Particularly an undercatch correction was not taken into account when constructing Table 1 and Fig. 2 in . This will be corrected.*

- Page 4 and Figure 1: "Golfplatz" in the Figure versus "Golf course" in the text. "SLF2" site is named "SLF" in the map. How were the boarders of the Dischma catchment defined (topography based from the model?). I would recommend some light, partly transparent background color for the names, to improve readability. I have to admit, I am not a fan of topographic maps as background, especially if the legend is missing. Any chance to replace it with a more generalized map?
  *Thank you for pointing out these inconsistencies in labelling; they have been resolved. Furthermore, we added a white, slightly transparent box behind the labels. Unfortunately, an illustrative map that is not a topographic map is not available. However, in order to increase readability, we switched to a less detailed map. See the new map below in Fig. 1 in this document. The Dischma catchment border is provided by the Swiss Federal Office of the Environment (FOEN). We plan to amend the manuscript at this point, and explain that model grid points with the center point inside the (sub-)catchment border polygon are considered being part of the (sub-)catchment.*

- Page 6, line 5 ff: Are the interpolations done for each time step?
  *Yes, as with the other parameters, the interpolation for precipitation is also done at every Alpine3D time step of 1 hour with the help of the MeteoIO library. This will be made clear in the revised manuscript.*

- Page 6, line 14: I do not think that "initialization" is the correct term. Is it not parameterization?
  *This sentence will be revised based on a suggestion by Reviewer 1. We now term it "soil properties".*

- Page 7, line 21. "sub-catchments" – so is this approach some kind of HRU approach?
  *Although it sounds similar to a HRU approach, a major difference is that the surface processes at every grid point inside a sub-catchment are explicitly resolved by the Alpine3D model, for example by taking into account variations in altitude, incoming solar radiation as a function of aspect and slope angle. It is only determined here which grid cell is draining to which sub-catchment and the residence time within the sub-catchment, based on terrain analysis only (and not soil properties, land use, etc.). We will revise the description of the coupling of Alpine3D to the streamflow model, hopefully adequately avoiding confusion with the HRU approach.*

- Page 7, line 33: Again, the soil moisture is calculated for the first 40 cm. Can you clarify its relation to the 30 cm stated before and after.

  *We will add Fig. 2 to the revised manuscript (see below), indicating the soil layering in the simulations, as well as the soil moisture measurement depths. The choice for 40 cm is motivated by the fact that the upper soil moisture measurements taken at 10 cm and 30 cm will more or less represent the upper 40 cm of the soil. The dielectric sensor 10HS for soil moisture used in this study measures approximately a volume of 1.32 l, as specified by the manufacturer. We will amend the manuscript at this point.*

- The definition of a rainfall event is a bit broad. Do you used mowing 12 h sum? What if a rainfall event is ended by falling below the 3mm thresholds criteria, but followed by a >10mm event again. Why do you choose a time window of 12 mm. Did you do any concentration time analysis?

  *Yes, a 12 h moving sum was used, we will specify this in the manuscript. In the case mentioned by the reviewer (rainfall falling below 3mm, but followed by a >10mm event), two events will be taken into the analysis. The time window of 12 hours was arbitrarily chosen, motivated by the fact that we aimed to select rather intense events. In total 168 rainfall events and 301 snowpack runoff events were selected (i.e, on average 16.8 and 30.1 events per year, respectively). The average duration of an event was 21.8 hrs (rainfall) and 20.9 hrs (snowpack runoff). On average, there are 6.8 days in between rainfall events, excluding the winter season. There are 1.3 days in between snowpack runoff events, excluding the summer season. We will add this information to the revised manuscript.*

- Page 8, line 14, and Figure 3. A comment on the vegetation growth (?) during summer would be nice.

  *Thank you for this suggestion, we will discuss this in the revised manuscript.*

- Page 9, line 27 ff. In my opinion, the r2 is not the appropriate statistical measure here, as it does not consider any systematic offsets/biases. The application of the RMSE or similar would be more fair. Furthermore, can you set your results in light of other models of soil moistures in alpine terrains? Also to show that your results are pretty good.

  *We are actually interested in to what extend the simulations are able to reproduce the variability in soil moisture. As the comparison of the two measured soil moisture sensors at a single station and single depth shows, often a bias is already present between both measurements. This suggests a bias in the sensors which could be resolved by recalibration of the sensors. We therefore do not necessarily want to express the existing bias in the statistical measure and we prefer to keep the results for $r^2$. Note that the existence of a bias can be clearly identified by readers by the soil moisture figures we show. We will clearly discuss the existence of a bias in the revised manuscript. Regarding the comment about citation of existing literature, the studies we are aware of that both simulate and measure soil moisture in alpine terrain are the studies by Gurtz et al. (2003); Rössler and Löffler (2010); Kumar et al. (2013); Pasolli et al. (2013); Brocca et al. (2013); Pellet et al. (2016). We will discuss our results in light with the results published in these studies.*

- Page, line 10: "however, ...." Isn't this finding clear and logical as you only consider "deeper" water fluxes

  *This is true, and we will rephrase this sentence.*

- I am looking forward to the revised manuscript.

  *Thank you.*

**References**

Brakensiek, D., and W. Rawls (1994), Soil containing rock fragments: effects on infiltration, *Catena*, *23*(1), 99–110, doi:10.1016/0341-8162(94)90056-6.

Brocca, L., A. Tarpanelli, T. Moramarco, F. Melone, S. Ratto, M. Cauduro, S. Ferraris, N. Berni, F. Ponziani, W. Wagner, and T. Melzer (2013), Soil moisture estimation in alpine catchments through modeling and satellite observations, *Vadose Zone J.*, *12*, –.

Comola, F., B. Schaefli, A. Rinaldo, and M. Lehning (2015), Thermodynamics in the hydrologic response: Travel time formulation and application to Alpine catchments, *Water Resour. Res.*, *51*(3), 1671–1687, doi: 10.1002/2014WR016228.

Federal Office for the Environment (FOEN) (2017), Dischmabach – Davos, Kriegsmatte 2327, *https://www.hydrodaten.admin.ch/de/2327.html*, URL retrieved at 2017-05-24.

Gurtz, J., M. Zappa, K. Jasper, H. Lang, M. Verbunt, A. Badoux, and T. Vitvar (2003), A comparative study in modelling runoff and its components in two mountainous catchments, *Hydrol. Proc.*, *17*(2), 297–311, doi:10.1002/hyp.1125.

Kumar, M., D. Marks, J. Dozier, M. Reba, and A. Winstral (2013), Evaluation of distributed hydrologic impacts of temperature-index and energy-based snow models, *Adv. Water Resour.*, *56*(0), 77–89, doi: 10.1016/j.advwatres.2013.03.006.

Pasolli, L., G. Bertoldi, S. D. Chiesa, G. Niedrist, U. Tappeiner, M. Zebisch, and C. Notarnicola (2013), Multi-source and multi-scale soil moisture dynamic modelling in mountain meadows, in *2013 IEEE International Geoscience and Remote Sensing Symposium - IGARSS*, pp. 763–766, doi:10.1109/IGARSS.2013.6721269.

Pellet, C., C. Hilbich, A. Marmy, and C. Hauck (2016), Soil moisture data for the validation of permafrost models using direct and indirect measurement approaches at three alpine sites, *Front. Earth Sci.*, *3*, 91, doi:10.3389/feart.2015.00091.

Rössler, O., and J. Löffler (2010), Potentials and limitations of modelling spatio-temporal patterns of soil moisture in a high mountain catchment using WaSiM-ETH, *Hydrol. Proc.*, *24*(15), 2182–2196, doi:10.1002/hyp.7663.

[Figure]

Figure 1: Topographical map of the simulated domain, showing the locations of the stations. IMIS stations are shown in black, IRKIS stations in red, SensorScope stations in green, SwissMetNet stations in blue and Weissfluhjoch in brown. The Dischma catchment and the gauging station measuring streamflow in the Dischmabach at the outlet of the Dischma catchment are shown in cyan. The inset shows the location of the simulation domain (red square) in Switzerland. Maps reproduced by permission of swisstopo (JA100118).

[Figure]

Figure 2: Soil layering as used in the Alpine3D model. The three water fluxes used to drive the streamflow model are shown in blue arrows. The soil moisture measurements are indicated by brown circles.

---

## Author Response (AR1)

Please find the original comments in regular, our original responses as published in the Online Discussion in *italics*, and the final changes made to the manuscript in **bold**. Page and line numbers in italics and bold text refer to the original and revised manuscript, respectively. A track-changed version of the manuscript can be found after the response to reviewers.

**1 Reviewer 1**

In the manuscript "Influence of snow surface processes on soil moisture dynamics and streamflow generation in alpine catchments", the authors present a comprehensive modeling study using the model Alpine3D which was complemented with new descriptions for simulating soil moisture and streamflow. The distributed model was forced using meteorological station data at several points and validated by means of snow depth, soil moisture, and runoff measurements.

**1.1 General Comments**

The manuscript presents a modeling study using a detailed set of modules and methods to tackle the challenge of simulating the hydrology in a complex mountainous catchment with a fully distributed, spatially highly resolved model. The focus lies on the simulation of snow depth, soil moisture and the respective interplay of precipitation, snow melt, and runoff dynamics. The study gives valuable insights in the involved hydrological processes. The manuscript is well elaborated and written and is technically of very high quality. I recommend its publication after minor revisions. Generally, the presented analysis is a bit incomplete because of the lack of a groundwater description in the model. This is mentioned in the manuscript at the respective sections. But it should be emphasized even more that this is a major zhortcoming of the study and it should be addressed in future work with the model setup. Another criticism is the description of the presented streamflow model. It is not quite clear to me how it was coupled to the model and what the flux input at or from different depths mean (sections 3.2 and 4.3). As I understand it, the water fluxes from the soil model at three different depths (lateral flux or excess out of the respective soil layer?) were taken and "streamed" into an external streamflow model. This streamflow model is calibrated using the respective fluxes which produces the shown streamflow simulations for three different depths. This approach is quite unusual and definitely needs further explanation in the manuscript. Why is the flux taken separately from the depths and not combined? The runoff dynamics clearly reveal that a groundwater module is missing. But this missing groundwater module could be "replaced" by a calibrated low flow component of the water flux (baseflow) which seems to be totally missing (Fig. 7, underestimated low flow / baseflow in the winter months). All other presented findings regarding soil moisture, freezing, as well as event-based precipitation and melt are well elaborated and very interesting. Some more questions that need clarification are listed in the following specific comments.

*We thank the reviewer for his positive remarks about the study and his constructive comments. We will take them into consideration when revising the manuscript. Regarding the ground water flow: here the issue is mainly that only the streamflow model treats groundwater flow, simulating the water storage dynamics of the deep soil compartment. However, there is no feedback to the Alpine3D model, such that a rising water table cannot be simulated in the Alpine3D model. One could envisage an approach where the level of the reservoirs in the streamflow model is coupled to the lower boundary of the SNOWPACK module in the Alpine3D model. However, it is not guaranteed that this approach will improve the soil moisture or streamflow simulations, as it also requires detailed information about soil properties and soil depths throughout the catchment. We will provide a more extensive discussion on ground water flow in the revised manuscript.*

*This comment is similar to the general comment of Reviewer 2 and for simplicity, we give the same response here: As both reviewers raise similar concerns, it is clear that we need to pay particular attention to describe the coupling between the Alpine3D and streamflow model in more detail and with more clarity when revising the manuscript. The streamflow model is a spatially explicit hydrologic response model at sub-catchment scale. Each sub-catchment is identified based on geomorphological analysis of the watershed. The model simulates the water storage dynamics in two soil compartments, namely an upper and lower one, of each sub-catchment using a travel time distribution approach. Outflow from the upper compartment represents interflow, while that from the lower component represents baseflow. We would like to point out that our model is reproducing baseflow, albeit too low at the end of the winter. If one would be particularly interested in correctly representing baseflow, a recalibration of the streamflow model with a focus on the statistics for the winter period would allow to have a more accurate representation of baseflow. But we do not agree with both reviewers that the baseflow is absent. Furthermore, the streamflow model needs a surface scheme, to provide the influx into the system. For this, we use the Alpine3D model. However, it is somehow arbitrary where to draw the boundary between the surface scheme and the streamflow model. For this, we tested 3 scenarios: a soil flux at 2, 30 and 60 cm depth. So we do not use the fluxes combined, but we used the three fluxes as three different scenarios. Although it would be similar as running 3 separate simulations, with either 2, 30 or 60 cm of soil, this approach would have the disadvantage that specifying the lower boundary condition for the Alpine3D model becomes tricky. For example, at 3 m depth, one can assign a constant geothermal heat flux and a water table and this would hardly influence the snowpack dynamics. On the other hand, at 2 cm below the surface, a constant geothermal heat flux would provide a too strong heating of the snowpack, as the soil buffer is not represented. Therefore, we choose the approach of doing a single simulation, and extracting soil water fluxes at three depths in order to test how to achieve an optimal coupling between the surface scheme Alpine3D and the streamflow model. This is illustrated in Fig. 3 in* Comola et al. (2015). *We will rewrite the Methods section discussing the model coupling thoroughly, in order to better explain our approach.*

**First, we exchanged the wording "streamflow model" for "hydrologic response model" throughout the manuscript, to emphasize that the model is not only treating streamflow, but the hydrologic response from the catchment, based on input water fluxes from the surface scheme provided by Alpine3D. We now state in the abstract that the groundwater description is missing in the Alpine3D model, see P.1, L.8. As we now explain in the revised section about the hydrologic response model, this model is describing inter- and groundwater flow, see P.8, L.7-L22. This section is also revised to better explain the coupling strategy between Alpine3D and the hydrologic response model.**

*Please find our response to other issues raised by the reviewer below.*

**1.2 Specific Comments**

- P.1 L. 6: "in close proximity to" instead of "in close proximity of"
  *Will be corrected, thank you.*
  **Corrected, see P.1, L.6.**

- P. 1 L. 9-15: "Streamflow simulations performed with a spatially-explicit hydrological model using a travel time distribution approach coupled to Alpine3D provided a closer agreement with observed streamflow at the outlet of the Dischma catchment when including 30 cm of soil layers. Performance decreased when including 2 cm or 60 cm of soil layers. This demonstrates that the role of soil moisture is important to take into account when understanding the relationship between both snowpack runoff and rainfall and catchment discharge in high alpine terrain." The differences in NSE for three simulations

are so small that I would not give this strong statement. It is also not at all an evidence for your second statement as you show no simulations without the new soil model, see comment below (P. 10 L. 18/19). It is for sure correct that soil moisture has to be taken into account but you show no real proof in this work.

*We agree with the Reviewer that the abstract was not accurately reflecting the results from our study at this point. Note that NSE coefficients are all very similar, as we recalibrated the streamflow model for each case individually. However, the conclusion about the importance of soil is not only drawn based on the NSE coefficients for discharge, but also for the relationship between initial soil moisture and runoff coefficients. We will rephrase this part of the abstract when revising the manuscript.*

**Corrected, see P.1, L.9-21.**

- P.1 L. 17: "which shows" instead of "and this shows"
*Will be corrected, thank you.*
**Corrected, see P.1, L.21.**

- P. 3 L. 7: Rephrase: "The measurement site Weissfluhjoch (WFJ), which is focused on snow-related measurements, as well as several permanent meteorological stations are located in close proximity to the area." instead of "The measurement site Weissfluhjoch (WFJ), which is focussed on snow-related measurements, is located in close proximity of the area, as well as several permanent meteorological stations."
*Will be rephrased, thank you.*
**Rephrased, see P.3, L.19-21.**

- P. 3 L. 14: "of total precipitation" instead of "of all precipitation"
*Will be corrected, thank you.*
**Corrected, see P.3, L.27.**

- P. 4 L. 7: Better use "focused", not "focussed" (see also above P. 3 L. 7)
*Will be corrected, thank you.*
**Corrected, see P.4, L.20.**

- P. 5 L. 14: Lower computational costs compared to what other approach? Please add an example for clarification!
*It was meant here: the bucket scheme for snow has a lower computation cost than the full Richards equation and the bucket scheme is an appropriate choice when the main interest is for seasonal and daily time scales. We will improve the wording of the manuscript at this point.*
**Corrected, see P.6, L.2.**

- P. 5 L. 21: Remove brackets in citation!
*Will be corrected, thank you.*
**Sentence has been rephrased, see P.6, L.3-4.**

- P. 6 L. 13: Rephrase the first two sentences / the beginning of this section ("Two important components to initialise Alpine3D simulations are the digital elevation model (DEM) for the Davos area, provided by the Swiss Federal Office of Topography (swisstopo). Also the soil has to be initialised for each pixel, although limited information is available.") e.g.: "Two important components to initialise Alpine3D simulations are the digital elevation model (DEM) and distributed soil information. The DEM is provided..."

*Thank you for the suggestion, this part will be rephrased.*
**Corrected, see P.6, L.30-32.**

- P. 7 L. 6: Either remove "on a computer cluster from 2008." or preferably provide some more information about the HPC system (e.g. type and clock speed of nodes). I guess the 14 hours per year using 36 CPU cores are the necessary wall clock time (or CPUh?). Please add this information in the manuscript.
  *We will revise the sentence as follows: "Using 36 CPU cores from a HPC system consisting of in total 32 compute nodes with two 6-core AMD Opteron 2439, 2.8 GHz processors per compute node, the computation took on average 14 hours wall clock time for a single year, mainly depending on the snow height in the winter season."*
  **Corrected, see P.7, L.25-27.**

- P. 7 L. 11: Remove "also".
  *Will be corrected, thank you.*
  **Corrected, see P.7, L.32.**

- P. 7 L. 12: Why didn't you additionally inspect hourly values if you have the respective measurements? You could add at least some examples for showing the model performance on a smaller, hourly timescale, which would be very interesting to see.
  *We agree with the reviewer that it is interesting to see the hourly behaviour of the soil moisture measurements and simulations. We therefore plan to amend the manuscript with an additional figure showing for one station an example of hourly soil moisture, during both the melt season as well as the summer season (see Fig. 1).*
  **The figure has been added as Fig. 7 and is now discussed in P.9, L.24-27 an P.9, L.29 - P.10, L.2.**

- P. 8 L. 6 ff and above and Figures 3–5: Consequently use one throughout the manuscript: either "snow depth" or "snow height" (personally, I prefer "depth").
  *We will use the term "snow depth" throughout the manuscript and in the figures.*
  **This has been corrected throughout the manuscript.**

- P. 8 L. 12 ff and Fig. 3: Please try to remove the measurement errors in Fig. 3 (high frequency fluctuations, especially in the summer months)! In June / July 2012 and 2013, the model seems to miss the measured spring snow fall at stations (a) and (b). Why does this happen? Add a respective explanation in the manuscript.
  *These measurement sites showing the high frequency fluctuations measure over a meadow and the snow depth measurements are not only recording the grass growth, but are also receiving a noisy signal from the grass. A few times during the summer, the grass below the snow depth sensor is mowed by farmers, which is also visible in the signal. We did not explain this in the original manuscript, but, as also suggested by Reviewer 2, we plan to explain the measured signal in the revised manuscript. We do not want to filter the signal, as it is a typical signal for grass growth and thereby recognizable as such.*
  **Explanation has been added in P.9, L.17-18.**

- P. 8 L. 15: To be consistent with the section title of 4.1 either remove "Measurements and Simulations" or add it in 4.1
  *Thank your for pointing out the inconsistency, we will shorten section title 4.2.*
  **Corrected, see P.9, L.19.**

- P. 9 L. 20: "S1" instead of "S3"

  *Will be corrected, thank you.*

  **Corrected, see P.11, L.3.**

- P. 9 L. 27: Are the r2 values calculated using the daily or hourly values? I guess daily, but please add this in the manuscript for clarification.

  *These concern hourly values. As the reviewer was expecting daily average values, we changed the graph accordingly (no significant change in results). We are sorry for causing confusion, but we will add this information in the manuscript.*

  **Corrected, see P.11, L.10.**

- P. 10 L. 12: Remove "us".

  *Will be corrected, thank you.*

  **Corrected, see P.12, L.2.**

- P. 10 L. 15: Please either explain your concept of the "virtual lysimeter" or use another notion! I think you are referring to the water fluxes at the three depths, but this is not clear here.

  *This indeed refers to the water fluxes at the three depths. We will rephrase this sentence and will replace the term "virtual lysimeter" with an explicit description.*

  **Rephrased, see P.12, L.4-5.**

- P. 10 L. 18/19: I am not sure if I understand this right, but the statement "The results suggests that the updated soil module of SNOWPACK is contributing to a better prediction of streamflow in the summer months." is misleading or drawn without any evidence. You show no Alpine3D runoff result without the new model, because – as I understand it – there was no soil moisture or runoff description in the model before. So a valid statement would be something like "The results show that the new soil module of SNOWPACK is enabling a simulation of streamflow."

  *We agree with the reviewer that the statement was misleading. There actually has been a very basic soil module in SNOWPACK for many years now, where water flow was described using a bucket-type approach. However, we did not want to aim for a comparison, as some very important physics is missing in the old module, for example water retention and water flow rates as a function of soil moisture. We think that a base-line soil model in a physics based model should at least apply Richards equation or something similar to describe water flow in soil. Nevertheless, taking soil water fluxes at 2 cm depth can be regarded as almost equivalent to directly routing snow melt and rainfall to the runoff routine (P7, L21-23), which we found to give a lower score than integrating 30 cm of soil layers. Our aim is to show that using a physics based description of soil processes improves the simulation of catchment discharge. We will rephrase parts of the manuscript to better explain our reasoning, which we think is in line with the suggestion by the reviewer.*

  **Rephrased, see P.12, L.8-11 and an improved description of the methodology is now present in P.8, L.8-22.**

- P. 11 last paragraph of section 4.4: The conclusions here are of course valid but were somehow clear before your study and should be underlined with existing literature.

  *We will refer to the appropriate literature in this section.*

  **The references has been added, see P.13, L.6-9.**

- Fig. 2, 3, 4, 5 and S1 – S5: Please add the year to the time-axis. This makes it much easier to look at when you write about single years in the text.

*This is a very good suggestion and the appropriate changes will be made.*

**Please find the updated figures in the revised manuscript as well as the updated Online Supplement.**

- Fig. 7, caption: typo "tics"

*Will be corrected, thank you.*

**Corrected, see caption of Fig. 9, which was previously Fig. 7.**

- Fig. 8, caption: I cannot see any data points plotted on the x-axis as stated in the caption. When you add them, please add the real value somehow because it is of interest how negative the NSE values are in these periods.

*Sorry for raising the confusion, but this remark referred to an earlier version of the plots, where NSE coefficients for summer 2012 were negative, due to, as was found later, a data processing error. This has been resolved. Now all NSE coefficients are positive and the manuscript will be updated accordingly.*

[Figure]

Figure 1: Measured and simulated soil moisture at the IRKIS station SLF2, for 10 cm depth (a, c) and 30 cm depth (b, d), during the snow melt season (a, b) and a snow-free summer month (c, d). In (a) simulated snow depth and in (c) precipitation is shown.

**2   Reviewer 2**

The manuscript mainly presents a comprehensive, multi-objective model validation study of an extension of the complex, physics-based Alpine 3D model. The model's performance to represent streamflow, snow depth, soil moisture and soil temperature/freezing is analyzed for 3 years in a Swiss mountain catchment and its surrounding. This evaluation is a framed by the context of the importance of soil moisture as a pre-disposition for floods. This framing – for my taste – is a bit wanted, and the title is somehow misleading. However, the quality of the model validation study is of a high standard and certainly of high importance. Especially the parallel evaluation of snow, soil and streamflow representation is cool. Although the overall impression of the article is very good, I do have some moderate remarks. The manuscript is certainly within the scope of HESS, it my knowledge of original content and can be a valuable article after my concerns are addressed.

*We thank the reviewer for the constructive comments and positive remarks about our study. We will take them into consideration when revising the manuscript. As we will discuss below, we agree that the title may be considered confusing and we propose a modification of the title. However, we would like to keep the framing of the study of soil moisture as a predisposition for flooding. This notion is introduced in the introduction section, with, in our opinion, appropriate citations. Furthermore, we do find evidence that our model framework is able to reproduce the relationship between soil saturation at the onset of large rainfall or snowmelt events and the discharge behaviour in the Dischma catchment.*

**2.1   General comments:**

- As mentioned before, I find the title a bit misleading, as it reads as the influence of snow processes on soil moisture and streamflow is analyzed. Instead, the focus is clearly on the model validation to reproduce the linkage between snow, soil and streamflow. I would recommend a rephrasing of the title.
  *We agree that the title is somewhat misleading, so we suggest to add the word "Simulating" in the beginning of the title, so we propose now: "Simulating the influence of snow surface processes on soil moisture dynamics and streamflow generation in alpine catchments".*
  **The title has been revised to: "Simulating the influence of snow surface processes on soil moisture dynamics and streamflow generation in an Alpine catchment".**

- Alike reviewer 1 (I haven't read his comments until I finished my review), I do not understand the usage of the soil water fluxes for streamflow generation. As reviewer 1 wrote to the point, why should one just take one of the fluxes (-2cm indicating surface runoff, -30 cm indicating interflow, -60 cm indicating baseflow). I also had a look in the cited publication, yet I find the entire concept very unusual and irritating. Because of this (I guess), the interpretation of the influence of soil moisture (Page 10, lines 14 ff) on streamflow is a bit simple. E.g. "neglecting the soil layers almost completely, by routing the 2 cm flux to the runoff model, is reducing the model efficiency". – This is logical as you neglect interflow and baseflow in summer months and interflow widely considered to be the dominant process in alpine catchments. Hence, the concept of this streamflow generation needs to be clarified and its strong limitation in terms of dynamic runoff generation should be discussed. Furthermore, effects of this simplified approach on the interpretation should be discussed.
  *This comment is similar to the general comment of Reviewer 2 and for simplicity, we give the same response here: As both reviewers raise similar concerns, it is clear that we need to pay particular attention to describe the coupling between the Alpine3D and streamflow model in more detail and with more clarity when revising the manuscript. The streamflow model is a spatially explicit hydrologic response model at sub-catchment scale. Each sub-catchment is identified based on geomorphological analysis of*

*the watershed. The model simulates the water storage dynamics in two soil compartments, namely an upper and lower one, of each sub-catchment using a travel time distribution approach. Outflow from the upper compartment represents interflow, while that from the lower component represents baseflow. We would like to point out that our model is reproducing baseflow, albeit too low at the end of the winter. If one would be particularly interested in correctly representing baseflow, a recalibration of the streamflow model with a focus on the statistics for the winter period would allow to have a more accurate representation of baseflow. But we do not agree with both reviewers that the baseflow is absent. Furthermore, the streamflow model needs a surface scheme, to provide the influx into the system. For this, we use the Alpine3D model. However, it is somehow arbitrary where to draw the boundary between the surface scheme and the streamflow model. For this, we tested 3 scenarios: a soil flux at 2, 30 and 60 cm depth. So we do not use the fluxes combined, but we used the three fluxes as three different scenarios. Although it would be similar as running 3 separate simulations, with either 2, 30 or 60 cm of soil, this approach would have the disadvantage that specifying the lower boundary condition for the Alpine3D model becomes tricky. For example, at 3 m depth, one can assign a constant geothermal heat flux and a water table and this would hardly influence the snowpack dynamics. On the other hand, at 2 cm below the surface, a constant geothermal heat flux would provide a too strong heating of the snowpack, as the soil buffer is not represented. Therefore, we choose the approach of doing a single simulation, and extracting soil water fluxes at three depths in order to test how to achieve an optimal coupling between the surface scheme Alpine3D and the streamflow model. This is illustrated in Fig. 3 in Comola et al. (2015). We will rewrite the Methods section discussing the model coupling thoroughly, in order to better explain our approach.*

**First, we exchanged the wording "streamflow model" for "hydrologic response model" throughout the manuscript, to emphasize that the model is not only treating streamflow, but the hydrologic response from the catchment, based on input water fluxes from the surface scheme provided by Alpine3D. We now state in the abstract that the groundwater description is missing in the Alpine3D model, see P.1, L.8. As we now explain in the revised section about the hydrologic response model, this model is describing inter- and groundwater flow, see P.8, L.7-L.22. This section is also revised to better explain the coupling strategy between Alpine3D and the hydrologic response model.**

*Please find our response to other issues raised by the reviewer below.*

- The description of the different soil layers is unclear: You introduced increasing soil layer magnitudes (from 2cm to 40 cm) up to a soils depth of 300 cm in the model. However, you take water fluxes from 2, 30 and 60 cm. Moreover you compare these to soil moisture measured at 10, 30, 50, 80, and 120 cm depth. And finally, you take the average of the upper 40 cm (page 10, line 25) as the soil moisture state within the catchment. As these number do not match at a first glance, a clarification is advisable. Maybe a sketch would help.

  *We think this is a very good suggestion and we will add a descriptive illustration in the revised manuscript. Please find the new figure below as Fig. 3. Note that the layer spacing was erroneously reported as ranging from 2 cm to 40 cm, where it should have been from 2 cm to 25 cm. This will be corrected in the revised manuscript.*

  **The figure has been included as Fig. 2 in the revised manuscript. Text in P.7, L.16 has been corrected.**

- In the manuscript, the SNOWPACK and Alpine3D are described as two separate models (e.g. in the model description and partly in the introduction). But as written in the Conclusion, SNOWPACK is

a module of Alpine3D and as I understand an integrated part of Alpine3D. This should be clarified throughout the text, especially in the beginning (Aims section)

*The reviewer is correct, the SNOWPACK model serves as the snow and soil module for the distributed Alpine3D model. When revising the manuscript, we will pay attention that this is correctly formulated throughout the manuscript.*

**This has been rephrased where necessary, see for example P.5, L.31-32.**

- As the Dischma catchment is an alpine catchment I assume that skeleton fraction is a major issue, both for measuring the "correct" soil moisture as well as for simulating the soil moist dynamics. Please, clarify how and if the skeleton fraction was considered in the pedo-transfer-function and how it was considered in the selection of the measuring location (and how representative the selection in terms of skeleton fraction is). Moreover, please discuss if the found biases in the soil moisture and soil temperature simulations can be explained by skeleton fraction. Finally (I hope I did not miss it), how do the soil types of all measuring stations represent the soil types in the Dischma catchment.

*- We agree that the skeleton fraction in alpine catchment is an important factor to take into account. For example, the study by Rössler and Löffler (2010) demonstrates in a sensitivity study that changing the skeleton fraction has an impact on streamflow and soil moisture simulations. They particularly describe the effect of an increase in porosity, and associated changes in hydraulic parameters. However, the study by Rössler and Löffler (2010) also points out that spatial variability of the skeleton fraction is largely unknown. In the current version of the SNOWPACK model, which provides the surface scheme for the Alpine3D model, the skeleton fraction is not taken into account, as the SNOWPACK uses prescribed soil types. We actually found that for some sites we get an adequate soil moisture simulation without considering the skeleton fraction, whereas for other sites the simulations are showing less agreement with measurements. But these contrasting results indicate that with the current information, only ad-hoc modifications of the skeleton fraction are possible, as we cannot separate well enough between the soil moisture sites based on available information (land use and soil permeability). For further development of the SNOWPACK and Alpine3D model, this certainly is an area of attention. As discussed by Brakensiek and Rawls (1994), neglecting rock fragments in soil may overestimate hydraulic conductivity. Thus, the bias in soil moisture we found can be explained by an overestimation of hydraulic conductivity in the model, which would bring down liquid water faster. As wetter soils need more energy to freeze, the underestimation of soil moisture in the top layer may also result from this bias. The above mentioned points will be discussed in the revised manuscript.*

**We discuss this now in P.7, L.11-14 and the possible effect on our simulations in P.10, L.17-23.**

*- The representativeness of the soil moisture measurement sites is given by that the Grossalp and Pischa stations were located in the "alpine meadow" class, which is 21.1% of the land use coverage (see Table 4 in the original manuscript). The Uf den Chaiseren, Dorfji and Stillberg stations are located in the "mixed forest", "bush" and "bare soil" class, respectively, which is found in 12.9%, 7.3% and 6.0% of the Dischma catchment, respectively. The SLF2 and Golf Course stations would officially fall into the category of "settlement", but one would describe the area as "alpine meadow". We will add this information to the manuscript.*

**Please find this information in P.5, L.13-18.**

- The description of the meteorological data is quite long and very detailed. I would suggest to just briefly describe the table 2.

*When revising the manuscript, we will put effort in shortening this section.*

**We shortened the section, see P.4, L.16 - P.5, L.28. However, we whish to keep a certain level of**

**detail, as some of the datasets collected in this research will be made publicly available via doi: 10.16904/17.**

**2.2 Specific comments:**

- Page 1, line 11, and 12: Please clarify the word "including", as you do not combine the three layers.
  *As our description of the coupling to the streamflow model was clearly confusing in the original manuscript, we plan to revise this part of the abstract as: "Streamflow simulations performed with a spatially-explicit hydrological model using a travel time distribution approach coupled to Alpine3D provided a closer agreement with observed streamflow at the outlet of the Dischma catchment when driving the streamflow model with soil water fluxes at 30 cm depth. Performance decreased when using the 2 cm soil water flux, thereby mostly ignoring soil processes. This demonstrates that the role of soil moisture is important to take into account when understanding the relationship between both snowpack runoff and rainfall and catchment discharge in high alpine terrain. However, using the soil water flux at 60 cm depth to drive the streamflow model also decreased its performance, indicating that an optimal soil depth to include in the simulations exists."*
  **Please find the revised abstract regarding this point in P.1, L9-17.**

- Page 2, line 29: "small scale surface processes". Please, specify the scale.
  *This refers to 10-100m scale, on which wind drifts form, and for which local topography strongly influences the energy balance via the slope aspect, angle and local shading. We will amend the manuscript at this point.*
  **Specified, see P.3, L.2-4.**

- Page 3 ,line 5: Please, specify the catchment size.
  *We will report that the catchment size that is represented by the gauging station is 43.3 km$^2$, as reported by the Swiss Federal Office of the Environment (Federal Office for the Environment (FOEN), 2017).*
  **Specified, see P.3, L.18.**

- Page 3, line 13 ff & Figure 2: How did you separated snow from rain here.
  *In the manuscript, we did all separations of precipitation in rain and snowfall based on an air temperature threshold of 1.2 °C for half-hourly measurements. We will specify this in the manuscript where necessary.*
  **Specified, see P.3, L.28-29.**

- Page 3, and Table 1: A comparison to the long term norm period would be interesting
  *We agree with this suggestion. We now add the 10-year averages to the table, which corresponds to the period for which the streamflow simulations were performed. Note that we came across an inconsistency. The data shown was not based on the same meteorological dataset as used for the Alpine3D simulations. Particularly an undercatch correction was not taken into account when constructing Table 1 and Fig. 2 in . This will be corrected.*
  **10 year averages were added to Table 1.**

- Page 4 and Figure 1: "Golfplatz" in the Figure versus "Golf course" in the text. "SLF2" site is named "SLF" in the map. How were the boarders of the Dischma catchment defined (topography based from the model?). I would recommend some light, partly transparent background color for the names, to improve readability. I have to admit, I am not a fan of topographic maps as background, especially if the legend is missing. Any chance to replace it with a more generalized map?

*Thank you for pointing out these inconsistencies in labelling; they have been resolved. Furthermore, we added a white, slightly transparent box behind the labels. Unfortunately, an illustrative map that is not a topographic map is not available. However, in order to increase readability, we switched to a less detailed map. See the new map below in Fig. 2 in this document. The Dischma catchment border is provided by the Swiss Federal Office of the Environment (FOEN). We plan to amend the manuscript at this point, and explain that model grid points with the center point inside the (sub-)catchment border polygon are considered being part of the (sub-)catchment.*

**Please find the updated map as Fig. 1 and the mapping of the catchment polygons on P.8, L.20-22 in the revised manuscript.**

- Page 6, line 5 ff: Are the interpolations done for each time step?

  *Yes, as with the other parameters, the interpolation for precipitation is also done at every Alpine3D time step of 1 hour with the help of the MeteoIO library. This will be made clear in the revised manuscript.*
  **Please see P.6, L.21-22.**

- Page 6, line 14: I do not think that "initialization" is the correct term. Is it not parameterization?

  *This sentence will be revised based on a suggestion by Reviewer 1. We now term it "soil properties".*
  **Corrected, see P.6, L.31-33.**

- Page 7, line 21. "sub-catchments" – so is this approach some kind of HRU approach?

  *Although it sounds similar to a HRU approach, a major difference is that the surface processes at every grid point inside a sub-catchment are explicitly resolved by the Alpine3D model, for example by taking into account variations in altitude, incoming solar radiation as a function of aspect and slope angle. It is only determined here which grid cell is draining to which sub-catchment and the residence time within the sub-catchment, based on terrain analysis only (and not soil properties, land use, etc.). We will revise the description of the coupling of Alpine3D to the streamflow model, hopefully adequately avoiding confusion with the HRU approach.*
  **Please find the revised description of the coupling of Alpine3D to the streamflow model on P.8, L.7-21.**

- Page 7, line 33: Again, the soil moisture is calculated for the first 40 cm. Can you clarify its relation to the 30 cm stated before and after.

  *We will add Fig. 3 to the revised manuscript (see below), indicating the soil layering in the simulations, as well as the soil moisture measurement depths. The choice for 40 cm is motivated by the fact that the upper soil moisture measurements taken at 10 cm and 30 cm will more or less represent the upper 40 cm of the soil. The dielectric sensor 10HS for soil moisture used in this study measures approximately a volume of 1.32 l, as specified by the manufacturer. We will amend the manuscript at this point.*
  **Please find the Figure as Figure 2 in the revised mansucript. The measurement volume of the sensors is listed on P.5, L20 and the explanation is added to P.8, L.33-34.**

- The definition of a rainfall event is a bit broad. Do you used mowing 12 h sum? What if a rainfall event is ended by falling below the 3mm thresholds criteria, but followed by a >10mm event again. Why do you choose a time window of 12 mm. Did you do any concentration time analysis?

  *Yes, a 12 h moving sum was used, we will specify this in the manuscript. In the case mentioned by the reviewer (rainfall falling below 3mm, but followed by a >10mm event), two events will be taken into the analysis. The time window of 12 hours was arbitrarily chosen, motivated by the fact that we aimed to select rather intense events. In total 168 rainfall events and 301 snowpack runoff events were selected*

*(i.e, on average 16.8 and 30.1 events per year, respectively). The average duration of an event was 21.8 hrs (rainfall) and 20.9 hrs (snowpack runoff). On average, there are 6.8 days in between rainfall events, excluding the winter season. There are 1.3 days in between snowpack runoff events, excluding the summer season. We will add this information to the revised manuscript.*
**Please find the added information on P.9, L.3-7.**

- Page 8, line 14, and Figure 3. A comment on the vegetation growth (?) during summer would be nice.
*Thank you for this suggestion, we will discuss this in the revised manuscript.*
**Please find the discussion of this point in the revised manuscript P.9, L.17-18.**

- Page 9, line 27 ff. In my opinion, the r2 is not the appropriate statistical measure here, as it does not consider any systematic offsets/biases. The application of the RMSE or similar would be more fair. Furthermore, can you set your results in light of other models of soil moistures in alpine terrains? Also to show that your results are pretty good.
*We are actually interested in to what extend the simulations are able to reproduce the variability in soil moisture. As the comparison of the two measured soil moisture sensors at a single station and single depth shows, often a bias is already present between both measurements. This suggests a bias in the sensors which could be resolved by recalibration of the sensors. We therefore do not necessarily want to express the existing bias in the statistical measure and we prefer to keep the results for $r^2$. Note that the existence of a bias can be clearly identified by readers by the soil moisture figures we show. We will clearly discuss the existence of a bias in the revised manuscript. Regarding the comment about citation of existing literature, the most important studies we are aware of that both simulate and measure soil moisture in alpine terrain are the studies by Gurtz et al. (2003); Rössler and Löffler (2010); Kumar et al. (2013); Pasolli et al. (2013); Brocca et al. (2013); Pellet et al. (2016). We will discuss our results in light with the results published in these studies.*
**Please find the discussion of this point in the revised manuscript P.11, L.20-25. Note that we explicitly mention the bias in the conclusions (P.13, L.21), including the bias between two measurements at the same depth at the same site (P.13, L.22).**

- Page, line 10: "however, ...." Isn't this finding clear and logical as you only consider "deeper" water fluxes
*This is true, and we will rephrase this sentence.*
**We amended this sentence, see P.11, L.32 - P.1, L.3.**

- I am looking forward to the revised manuscript.
*Thank you.*

**References**

[revised manuscript text omitted]